# Causality and the Interpretation of Quantum Mechanics

Kaixun Tu[1, *]    Qing Wang[1, 2, †]

[1] *Department of Physics, Tsinghua University, Beijing 100084, China*

[2] *Center for High Energy Physics, Tsinghua University, Beijing 100084, China*

**Abstract**

From the ancient Einstein-Podolsky-Rosen paradox to the recent Sorkin-type impossible measurements problem, the contradictions between relativistic causality, quantum non-locality, and quantum measurement have persisted. Unlike recent approaches that address causality by enlarging the Hilbert space to introduce the quantum states of the detectors, our perspective is that everything—including detectors, the environment, and even humans—is composed of the same fundamental fields. This implies that the Hilbert space of the entire system remains the same regardless of the number of detectors. We employ reduced density matrices to characterize the local information of quantum states and show that reduced density matrices cannot evolve superluminally, thereby establishing causality. We further analyze the constraint that causality imposes on physical operations. Based on this constraint and the inherent entanglement in quantum field theory, we clarify that the traditional derivation of the Schrödinger's cat paradox is problematic, and we show that the Reeh–Schlieder theorem makes it possible that the Schrödinger's cat paradox does not arise. We then point out that, if the Schrödinger's cat paradox is indeed absent from the outset, this opens the door to an interpretation of quantum mechanics that respects causality and preserves the completeness of quantum mechanics without invoking the many-worlds scenario.

---

*Email: tkx19@tsinghua.org.cn

†Email: wangq@mail.tsinghua.edu.cn

# 1 Introduction

To discuss relativistic causality within the framework of quantum field theory, we need the concept of localization. In the case of single-particle states, one can attempt to discuss the localization of quantum states by defining position operators [1]. However, as succinctly argued in [2], this approach leads to superluminal propagation, indicating that position operators are not suitable for a genuine description of local phenomena in quantum field theory. Further variations and physical implications are discussed in [3–12]. Compared to superluminal propagation of particles, the "Fermi two-atom problem" can more accurately reflect the core of causality [13]. Fermi considered two atoms separated by a distance $R$, with one atom in the ground state and the other in the excited state. If causal influences propagate at the maximum speed $c$, then within the time interval $0 < t < R/c$, the excited atom will not have any impact on the ground-state atom through emitted photons.

The Fermi two-atom problem garnered renewed attention when it was revisited sixty years later by Hegerfeldt [14] (The introduction in [15] is a brief review of earlier work). Hegerfeldt fixed the local properties of states with the help of projection operators, and demonstrated that analyzing the Fermi two-atom problem in terms of transition probabilities leads to results that violate causality. Subsequently, Buchholz and Yngvason used algebraic quantum field theory to analyze the problem, and pointed out that Hegerfeldt's projection operators are not legitimate quantities and expectation values should be used instead. [16]. In the modern development of the Fermi two-atom problem, the two atoms were replaced by two Unruh-DeWitt detectors to explore causality [17–24]. In this context, detectors are not composed of fields used to propagate influences over the distances; instead, one needs to enlarge the Hilbert space to introduce the quantum states of the detectors. Additionally, the initial states are bare states (direct product states), involving the projection operators required by Hegerfeldt [14], which, however, was precisely the point refuted by Buchholz and Yngvason [16].

In the framework of quantum field theory, everything in the world is composed of the same fundamental fields, which means that we do not need to enlarge the Hilbert space to introduce detectors. More specifically, detectors are composed of elementary particles, and elementary particles are excitations of fundamental fields; therefore, detectors are essentially excitations of fundamental fields as well. Regardless of the number of detectors, the Hilbert space used to describe the entire system should remain the same. At the same time, this also implies that we cannot separate detectors from the systems being detected. Even if we consider quantum fields at different spatial points as different field operators, the interactions between fields at adjacent points lead to strong entanglement between different points in the quantum state. This also means that even the vacuum, such a basic and simple state, cannot be written as a direct product of "vacuum states" of different regions. The characteristic of quantum field theory has been noted by Hegerfeldt (see [14], last page) and further analyzed by Buchholz and Yngvason [16]. However, subsequent studies and discussions on causality have overlooked this characteristic and simply regarded the initial state as a bare state. Later, we will point out that the idea that "everything is composed of the same fundamental fields" plays an important role in finding an interpretation of quantum mechanics compatible with relativity.

In addition to the two-detector system mentioned above, introducing a third detector will lead

to "Sorkin-type impossible measurements problem" illustrating that the natural generalization of the non-relativistic measurement scheme to relativistic quantum theory leads to superluminal signaling [25–27]. Traditional quantum mechanics is governed by two laws: time evolution of quantum states and measurement theory. Relativity requires that these two laws together must ensure that information cannot propagate faster than light. However, non-relativistic measurement theories with traditional state update rule (Lüders' rule) cannot meet this requirement. To address this issue, various "update rules" have recently been proposed to describe the state update induced by measurements on the fields or on the detectors [26–32]. In fact, the Sorkin-type impossible measurements problem reveals a deeper issue concerning the very nature of measurement, providing a window for exploring the interpretation of quantum mechanics. Since the physical laws underlying measurement theory are tied to interpretations of quantum mechanics, and traditional non-relativistic measurement theory has been shown to be problematic, this suggests that traditional interpretations may also be flawed. A correct interpretation of quantum mechanics has the potential to provide a measurement theory that satisfies causality.

In this paper, we adopt a realistic perspective to investigate causality, where everything is composed of the same fundamental fields and collectively described by a pure state. This implies that there is no need to enlarge the Hilbert space to introduce detectors, and the Hilbert space of the entire system remains the same regardless of the number of detectors. Specifically, we employ reduced density matrices to characterize the local information of quantum states. Dividing the entire space into two regions, denoted as $a$ and $A$, we trace out region $a$ ($A$) to obtain the reduced density matrix which characterizes the information of the quantum state in region $A$ ($a$). As shown in Fig. 1, region $b$ is the intersection of the time slice $t$ with the causal future of region $a$. Consider two quantum states whose reduced density matrices in region $a$ are different, but whose reduced density matrices in region $A$ are identical at the initial time. In quantum field theory, we will show that their reduced density matrices in region $B$ at time $t$ must also be identical, thereby establishing causality.

The most important result of our paper is that relativistic causality, together with the idea that everything is composed of the same fundamental fields, implies that the traditional derivation of Schrödinger's cat paradox is problematic. The standard derivation of the Schrödinger's cat paradox [34] assumes that, prior to the measurement, the detector and the microscopic system being measured are unentangled, i.e., the joint state is of the form$|$System$\rangle|$Detector$\rangle$. However, since detectors and the systems under measurement are both composed of the same fundamental fields, the composite system formed by a detector and the system being measured cannot be written in this direct product form. In fact, even the vacuum state cannot be factorized into two unentangled spatial regions. Instead, the entire configuration must be treated as a global quantum field state. Nevertheless, the detector and the system being measured can be effectively recovered and characterized by their corresponding reduced density matrices. Moreover, since physical operations must obey fundamental physical laws, relativistic causality constrains our ability to manipulate the entire system. The pre-existing entanglement between the detector and the microscopic system under measurement, together with the causal constraint on physical operations, prevents us from deriving the Schrödinger's cat paradox within the framework of quantum field theory in the same way as in the traditional derivation.

In addition, we show that the Reeh–Schlieder theorem implies that a measurement outcome

may display only a single classical result, even when the quantum state corresponding to this classical outcome is not orthogonal to the quantum states corresponding to other possible classical outcomes. In other words, even if the final state of the measurement process is a superposition of macroscopically distinct classical configurations, the observed measurement outcome may still be a definite macroscopic state rather than a Schrödinger's cat–like superposition. This further supports the possibility that the Schrödinger's cat paradox may not arise in quantum field theory.

If the Schrödinger's cat paradox truly does not exist, then our understanding of the foundations of quantum mechanics would be fundamentally altered. The Schrödinger's cat paradox indicates that a superposition of microscopic states would lead to a superposition of distinct macroscopic detector readouts following the measurement process. Its essence is to show that, if quantum mechanics is complete, then microscopic superpositions would evolve into macroscopic superpositions through unitary evolution, which is at odds with actual measurement experiments. In other words, unitary evolution cannot turn possible results into actual results, yet real measurement experiments always display only a single outcome. Resolving Schrödinger's cat paradox is the central task of interpretations of quantum mechanics. The most popular interpretation is the Copenhagen interpretation, in which the Heisenberg cut restricts the domain of applicability of quantum mechanics and thus prevents the entire universe from being described by a single quantum state. Another well-known interpretation is the many-worlds interpretation, in which all possible measurement outcomes are real but belong to different worlds. Other proposals include Wigner's consciousness-induced collapse theory, Penrose's gravity-induced collapse theory, Bohm's hidden-variable theory, and so on (for an overview of the various interpretations of quantum mechanics, see Ref. [33]). Therefore, one can imagine that if the Schrödinger's cat paradox — which has been the root cause behind the emergence of the various interpretations of quantum mechanics — were in fact absent from the outset, our understanding of the foundations of quantum mechanics would be fundamentally altered. One important consequence is that we might obtain an interpretation that preserves the completeness of quantum mechanics without invoking the "many-worlds scenario". For convenience, we will figuratively refer to such a potential interpretation as the "one-world interpretation".

This paper is organized as follows. In Section 2, we demonstrate causality using the example of a free scalar field, providing a detailed exposition of our definition of causality. Subsequently, we extend the proof of causality to the general case of quantum field theory. In Section 3, we analyze the constraint that causality imposes on physical operations. Here, the term "physical operation" does not refer to the measurement process itself, but rather to the operations by which one prepares the microscopic system to be measured in a measurement experiment. In Section 4, we first briefly review the traditional derivation of the Schrödinger's cat paradox. We then discuss in detail the loopholes in this derivation and point out that these loopholes allow for a possible new interpretation of quantum mechanics that supports the completeness of quantum mechanics without invoking the many-worlds scenario. In addition, we show that the Reeh–Schlieder theorem further supports the possibility that the Schrödinger's cat paradox may not arise in quantum field theory, thereby enhancing the plausibility of the new interpretation of quantum mechanics. We also discuss how the new interpretation differs from traditional hidden-variable theories and analyze its compatibility with relativistic causality. Section 5 concludes the paper with a summary and some discussions, along with several directions for further investigations.

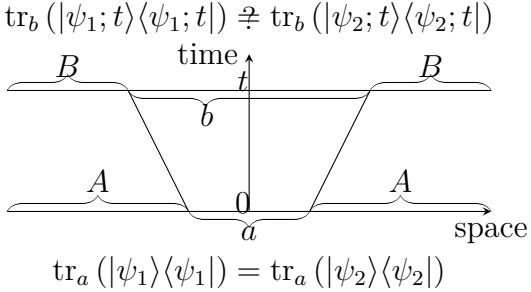

$$\mathrm{tr}_b\left(|\psi_1;t\rangle\langle\psi_1;t|\right) \overset{?}{=} \mathrm{tr}_b\left(|\psi_2;t\rangle\langle\psi_2;t|\right)$$

$$\mathrm{tr}_a\left(|\psi_1\rangle\langle\psi_1|\right) = \mathrm{tr}_a\left(|\psi_2\rangle\langle\psi_2|\right)$$

Figure 1 . The relationships among regions $A$, $a$, $B$, and $b$

## 2 Localization and Causality

In order to clearly present our definition of causality, we first use the free real scalar field as an explicit example to illustrate and carefully prove causality. At the same time, we introduce the notion of reduced density matrices in quantum field theory, which plays an important role in the later discussion of the Schrödinger's cat paradox. Subsequently, we extend the proof of causality to general local quantum field theories.

### 2.1 Free field theory as an example

Consider a free field theory described by the Hamiltonian $\hat{H} = \int \mathrm{d}^3x\left[\frac{1}{2}\hat{\pi}^2(\boldsymbol{x}) + \frac{1}{2}\left(\nabla\hat{\phi}(\boldsymbol{x})\right)^2 + \frac{1}{2}m^2\hat{\phi}^2(\boldsymbol{x})\right]$, and a quantum state $|\psi;t\rangle = \mathrm{e}^{-i\hat{H}t}|\psi\rangle$ evolving from an initial state $|\psi\rangle$ after a time $t$. In fact, the expression of $H$ in terms of creation and annihilation operators is precisely the same as $\hat{H}_{rS}$ in equation (3.3a) of Ref. [8]. In Ref. [8], $\hat{H}_{rS}$ is referred to as the "relativistic Schrödinger system", and intriguingly the authors claim that this theory would violate causality. However, we of course do not agree with that conclusion.

To highlight the local information of the quantum state, we use a representation based on the eigenstates $|\phi\rangle$ of the field operator $\hat{\phi}$, rather than the traditional Fock representation. These states are normalized according to $\langle\phi|\phi'\rangle = \prod_{\boldsymbol{x}}\delta(\phi(\boldsymbol{x}) - \phi'(\boldsymbol{x}))$. Sometimes, we interchange the symbol $\phi$ with $\varphi$. Utilizing the field propagator derived in Appendix A, the density matrix $\hat{\rho}(t) = |\psi;t\rangle\langle\psi;t|$ can be expressed as

$$\begin{aligned}
\rho(\phi,\phi';t) &= \langle\phi|\psi;t\rangle\langle\psi;t|\phi'\rangle \\
&= |N(t)|^2 \int \mathcal{D}\varphi\mathcal{D}\varphi'\,\rho(\varphi,\varphi')K(\phi,\phi',\varphi,\varphi';t)\,,
\end{aligned} \tag{1}$$

where

$$\begin{aligned}
\rho(\varphi,\varphi') &= \langle\varphi|\psi\rangle\langle\psi|\varphi'\rangle\,, &(2)\\
N(t) &= \mathcal{N}\mathrm{e}^{-\frac{1}{2}\int \mathrm{d}^3x \int \mathrm{d}t\, G(\mathbf{0};t)}\,, &(3)\\
K(\phi,\phi',\varphi,\varphi';t) &= \mathrm{e}^{iS(\phi,\varphi;t)-iS(\phi',\varphi';t)}\,, &(4)
\end{aligned}$$

$$S(\phi, \varphi; t) = \frac{1}{2} \int d^3x d^3y \, G(\boldsymbol{x} - \boldsymbol{y}; t) \Big[ \phi(\boldsymbol{x})\phi(\boldsymbol{y}) + \varphi(\boldsymbol{x})\varphi(\boldsymbol{y}) \Big]$$
$$- \int d^3x d^3y \, g(\boldsymbol{x} - \boldsymbol{y}; t)\phi(\boldsymbol{x})\varphi(\boldsymbol{y}) , \tag{5}$$

and

$$G(\boldsymbol{x} - \boldsymbol{y}; t) = \int \frac{d^3p}{(2\pi)^3} \frac{p^0 \cos(p^0 t)}{\sin(p^0 t)} e^{i\boldsymbol{p}\cdot(\boldsymbol{x}-\boldsymbol{y})} ,$$
$$g(\boldsymbol{x} - \boldsymbol{y}; t) = \int \frac{d^3p}{(2\pi)^3} \frac{p^0}{\sin(p^0 t)} e^{i\boldsymbol{p}\cdot(\boldsymbol{x}-\boldsymbol{y})} . \tag{6}$$

Dividing the entire space into two regions denoted as $a$ and $A$, we trace out region $a$ to obtain the reduced density matrix which characterizes the information of the quantum state in region $A$, namely $\int \mathcal{D}\varphi_a \, \rho(\varphi, \varphi')\Big|_{\varphi_a=\varphi'_a}$. As shown in Fig. 1, region $b$ is the intersection of the time slice $t$ with the causal future of region $a$. We trace out region $b$ to obtain the reduced density matrix which characterizes the information of the quantum state in region $B$, namely $\int \mathcal{D}\phi_b \, \rho(\phi, \phi'; t)\Big|_{\phi_b=\phi'_b}$. Using the derivation presented in Appendix B, we can obtain the relation between the above two reduced density matrices:

$$\int \mathcal{D}\phi_b \, \rho(\phi, \phi'; t)\Big|_{\phi_b=\phi'_b} = \int \mathcal{D}\varphi_A \mathcal{D}\varphi'_A F(\phi_B, \phi'_B, \varphi_A, \varphi'_A; t) \int \mathcal{D}\varphi_a \, \rho(\varphi, \varphi')\Big|_{\varphi_a=\varphi'_a}. \tag{7}$$

This indicates that the reduced density matrix of region $B$ at time $t$ in Fig. 1 is determined solely by the reduced density matrix of region $A$ at time $t = 0$. Therefore, if we consider two quantum states that have different reduced density matrices in region $a$ but identical reduced density matrices in region $A$, Eq. (7) implies that the reduced density matrices of region $B$ for the two quantum states are identical at time $t$.

If the quantum state $|\psi\rangle$ describes a group of particles (which may not be an eigenstate of the particle number) confined in region $a$ at $t = 0$, the quantum state in region $A$ is indistinguishable from the vacuum $|\Omega\rangle$, i.e., $\text{tr}_a(|\psi\rangle\langle\psi|) = \text{tr}_a(|\Omega\rangle\langle\Omega|)$. According to the established causality, for $|\psi; t\rangle = e^{-i\hat{H}t}|\psi\rangle$, we have $\text{tr}_b(|\psi; t\rangle\langle\psi; t|) = \text{tr}_b(|\Omega\rangle\langle\Omega|)$, which means that at time $t$ the quantum state in region $B$ remains indistinguishable from the vacuum state, indicating that the propagation speed of particles does not exceed the speed of light.

The purpose of writing this subsection together with Appendix B is not to provide a rigorous proof of causality, but rather to introduce the notion of reduced density matrices in quantum field theory, to demonstrate how one can work with them, and to illustrate the form that causality takes in this language. Our actual proof of causality is given in the following subsection, which is much shorter and simpler.

## 2.2   General cases

Ref. [35] demonstrated that: For a quantum field theory that can be expressed in the form $\hat{H} = \int d^3x \, \hat{h}(\boldsymbol{x}, t)$, the reduced density matrix of spacelike-separated regions will not be affected by the action of a human, if "the action of a human on quantities defined at some point of

coordinates $x$ and $t$ results only in changes $\Delta h(x,t)$ of the Hamiltonian density operator $h(x,t)$ defined at the same point" (refer to the third hypothesis in [35]). Ignoring the physical aspects and focusing solely on the mathematical expression, we can rephrase the mathematical conclusion in the aforementioned demonstration of Ref. [35] using the language of our paper as follows: If there exists a unitary operator $\hat{U}_a$ supported in region $a$ such that $|\psi_2\rangle = \hat{U}_a|\psi_1\rangle$ and the relationship between regions $a$ and $b$ is as shown in Fig. 1, then we have $\mathrm{tr}_b\left(|\psi_1;t\rangle\langle\psi_1;t|\right) = \mathrm{tr}_b\left(|\psi_2;t\rangle\langle\psi_2;t|\right)$.

Next, we will prove that if two quantum states $|\psi_1\rangle$ and $|\psi_2\rangle$ satisfy $\mathrm{tr}_a\left(|\psi_1\rangle\langle\psi_1|\right) = \mathrm{tr}_a\left(|\psi_2\rangle\langle\psi_2|\right)$, then there must exist a unitary operator $\hat{U}_a$ supported in region $a$ such that $|\psi_2\rangle = \hat{U}_a|\psi_1\rangle$. It is worth noting that this fact indeed appears as Exercise 2.81 in the well-known quantum information textbook by Nielsen and Chuang [36]. In this sense, our derivation here amounts to explicitly working out that exercise.

Denoting the basis of the Hilbert space in regions $A$ and $a$ as $|A_n\rangle$ and $|a_m\rangle$ respectively, the quantum states $|\psi_2\rangle$ and $|\psi_1\rangle$ can be expressed as

$$|\psi_i\rangle = \sum_{m=1}^{M}\sum_{n=1}^{N} f_i(m,n)|a_m\rangle|A_n\rangle\,, \quad i = 1,2 \tag{8}$$

Let $\boldsymbol{f}_i$ denote an $M \times N$ matrix where $f_i(m,n)$ is the matrix element at the $m$-th row and $n$-th column, then the condition $\mathrm{tr}_a\left(|\psi_1\rangle\langle\psi_1|\right) = \mathrm{tr}_a\left(|\psi_2\rangle\langle\psi_2|\right)$ can be expressed as

$$\boldsymbol{f}_1^\dagger\boldsymbol{f}_1 = \boldsymbol{f}_2^\dagger\boldsymbol{f}_2\,. \tag{9}$$

Suppose the rank of the $\boldsymbol{f}_2$ is $k = M$. For the case where $N = M$, $\boldsymbol{f}_2$ has an inverse and then we have $\boldsymbol{f}_2 = (\boldsymbol{f}_2^\dagger)^{-1}\boldsymbol{f}_1^\dagger\boldsymbol{f}_1$ from Eq. (9). Consequently, a unitary matrix $\boldsymbol{U} \equiv (\boldsymbol{f}_2^\dagger)^{-1}\boldsymbol{f}_1^\dagger$ can be defined such that $\boldsymbol{f}_2 = \boldsymbol{U}\boldsymbol{f}_1$. For the case where $N > M$, it is possible to select $M$ linearly independent column vectors from $\boldsymbol{f}_2$. Then we can form an $M \times M$ square matrix $\boldsymbol{F}_2$ using these $M$ linearly independent vectors, and it is evident that $\boldsymbol{F}_2$ is an invertible matrix. Based on the positions of the $M$ linearly independent vectors in $\boldsymbol{f}_2$, we can also correspondingly select column vectors from $\boldsymbol{f}_1$ to form another $M \times M$ square matrix $\boldsymbol{F}_1$. According to the definitions of $\boldsymbol{F}_1$ and $\boldsymbol{F}_2$, Eq. (9) immediately yields $\boldsymbol{F}_1^\dagger\boldsymbol{f}_1 = \boldsymbol{F}_2^\dagger\boldsymbol{f}_2$. Therefore, a unitary matrix $\boldsymbol{U} \equiv (\boldsymbol{F}_2^\dagger)^{-1}\boldsymbol{F}_1^\dagger$ can be defined such that $\boldsymbol{f}_2 = \boldsymbol{U}\boldsymbol{f}_1$.

Suppose the rank of the $\boldsymbol{f}_2$ is $k < M$. According to basic linear algebra knowledge, we know that $\mathrm{rank}\boldsymbol{f}_1 = \mathrm{rank}(\boldsymbol{f}_1^\dagger\boldsymbol{f}_1)$ and $\mathrm{rank}\boldsymbol{f}_2 = \mathrm{rank}(\boldsymbol{f}_2^\dagger\boldsymbol{f}_2)$. Thus, Eq. (9) indicates that the rank of $\boldsymbol{f}_1$ is $\mathrm{rank}\boldsymbol{f}_1 = \mathrm{rank}\boldsymbol{f}_2 = k$. It is possible to add $M - k$ normalized and mutually orthogonal column vectors to $\boldsymbol{f}_1$ ($\boldsymbol{f}_2$), such that these newly added vectors are orthogonal to the original ones in $\boldsymbol{f}_1$ ($\boldsymbol{f}_2$). After adding these vectors, $\boldsymbol{f}_1$ and $\boldsymbol{f}_2$ become extended matrices with rank $M$, denoted as $\boldsymbol{f}_1'$ and $\boldsymbol{f}_2'$. Due to the orthogonality and normalization of the new column vectors, as well as their orthogonality with the old column vectors, it is evident that (9) can be extended to $\boldsymbol{f}_1'^\dagger\boldsymbol{f}_1' = \boldsymbol{f}_2'^\dagger\boldsymbol{f}_2'$. Consequently, according to the proof in the preceding paragraph, there exists a unitary matrix $\boldsymbol{U}$ such that $\boldsymbol{f}_2' = \boldsymbol{U}\boldsymbol{f}_1'$. Focusing only on the transformation of the old column vectors, we immediately have $\boldsymbol{f}_2 = \boldsymbol{U}\boldsymbol{f}_1$.

In summary, we can always find a unitary matrix $\boldsymbol{U}$ such that $\boldsymbol{f}_2 = \boldsymbol{U}\boldsymbol{f}_1$. Let $U(m,m')$ be the matrix element of $\boldsymbol{U}$ at the $m$-th row and $m'$-th column. Using Eq. (8) and $\boldsymbol{f}_2 = \boldsymbol{U}\boldsymbol{f}_1$, we

obtain

$$
\begin{aligned}
|\psi_2\rangle &= \sum_{m=1}^{M}\sum_{n=1}^{N} f_2(m,n)|a_m\rangle|A_n\rangle \\
&= \sum_{m=1}^{M}\sum_{n=1}^{N}\sum_{m'=1}^{M} U(m,m')f_1(m',n)|a_m\rangle|A_n\rangle \qquad (10) \\
&= \left(\sum_{m=1}^{M}\sum_{m'=1}^{M} U(m,m')|a_m\rangle\langle a_{m'}|\right)|\psi_1\rangle \ .
\end{aligned}
$$

Eq. (10) indicates that there exist a unitary operator $\hat{U}_a$ supported in region $a$ such that $|\psi_2\rangle = \hat{U}_a|\psi_1\rangle$. As mentioned at the beginning of this section, combining this with the results from Ref. [35], we can establish causality: In Fig. 1, if two quantum states $|\psi_1\rangle$ and $|\psi_2\rangle$ initially satisfy $\mathrm{tr}_a\left(|\psi_1\rangle\langle\psi_1|\right) = \mathrm{tr}_a\left(|\psi_2\rangle\langle\psi_2|\right)$, then after time $t$, the two quantum states satisfy $\mathrm{tr}_b\left(|\psi_1;t\rangle\langle\psi_1;t|\right) = \mathrm{tr}_b\left(|\psi_2;t\rangle\langle\psi_2;t|\right)$.

It should be emphasized here that the unitary operator $\hat{U}_a$ constructed formally in Eq. (10) has no physical significance and it merely serves as an intermediate mathematical tool used to prove causality.

# 3  Physical Operations

The idea that everything is composed of the same fundamental fields implies that even the behavior of a human being, which may appear to involve free will, ultimately obeys the basic laws of physics. Consequently, the physical operations performed by an apparatus (or by a human) on a system are constrained by causality. The notion of a physical operation will be used later when we discuss measurement and the Schrödinger's cat paradox in Section 4. It is important to emphasize here that the role played by physical operations in a measurement experiment is not the measurement process itself, but rather the operations by which one prepares the microscopic system to be measured. Accordingly, the "apparatus (or a human)" referred to in this section denotes the means by which the initial states of an experiment are prepared, not the measuring instruments. Devices used for measurement are explicitly referred to as "detectors" in this paper, not as "apparatus".

To discuss more deeply how physical operations are constrained by causality, let us analyze Witten's remarks on physical operations in his discussion of reconciling causality with the Reeh–Schlieder theorem [37]: "If one asks about not mathematical operations in Hilbert space but physical operations that are possible in the real world, then the only physical way that one can modify a quantum state is by perturbing the Hamiltonian by which it evolves, thus bringing about a unitary transformation".

Consider two quantum states: the state $|\psi_1\rangle$ describes only the system being operated on, whereas the other state $|\psi_2\rangle$ includes not only that same system but also an apparatus (or a human) localized in region $a$. At time $t=0$, the apparatus in region $a$ has not yet performed any operation on the system, so the two quantum states are identical outside region $a$, specifically satisfying $\mathrm{tr}_a\left(|\psi_1\rangle\langle\psi_1|\right) = \mathrm{tr}_a\left(|\psi_2\rangle\langle\psi_2|\right)$. The apparatus complete the operation after $\Delta t$, and the

two quantum states evolve into $|\psi_1; \Delta t\rangle$ and $|\psi_2; \Delta t\rangle$, respectively. Combining with Fig. 1 (where $t = \Delta t$), causality ensures that $\mathrm{tr}_b\left(|\psi_1; \Delta t\rangle\langle\psi_1; \Delta t|\right) = \mathrm{tr}_b\left(|\psi_2; \Delta t\rangle\langle\psi_2; \Delta t|\right)$. Based on what we have proven in Section 2.2, this indicate the existence of a unitary operator $\hat{U}_b$ supported in region $b$ such that $|\psi_2; \Delta t\rangle = \hat{U}_b|\psi_1; \Delta t\rangle$.

However, the physical operation $\hat{U}_b$ defined above encompasses not only the action of the apparatus on the system being operated upon (i.e., the process $|\psi_2\rangle \to |\psi_2; \Delta t\rangle$) but also the transition from the state without the apparatus to the state with the apparatus (i.e., the process $|\psi_1\rangle \to |\psi_2\rangle$). If we wish to make $|\psi_2\rangle$ more similar to $|\psi_1\rangle$, we may let $|\psi_1\rangle$ also include the same apparatus but with it switched off, i.e., it will not perform any operation. We denote this new state as $|\psi_1'\rangle$. By carrying out a similar derivation, we can likewise obtain a unitary operator $\hat{U}_b'$ supported in region $b$ such that $|\psi_2; \Delta t\rangle = \hat{U}_b'|\psi_1'; \Delta t\rangle$. But even if $|\psi_1'\rangle$ includes the same apparatus, $\hat{U}_b'$ still incorporates the change of the apparatus from the "off" state to the "on" state.

If we wish the physical operation $\hat{V}_b$ to include only the action of the apparatus (i.e., $|\psi_2; \Delta t\rangle = \hat{V}_b|\psi_2\rangle$), then we must require that the system being operated on, $|\psi_1\rangle$, be an eigenstate of the Hamiltonian $\hat{H}$. In other words, in the absence of the operating apparatus, the system $|\psi_1\rangle$ remains unchanged under time evolution (e.g., Witten chose $|\psi_1\rangle$ to be the vacuum state in Ref. [37]). In this case, $|\psi_1; \Delta t\rangle\langle\psi_1; \Delta t| = |\psi_1\rangle\langle\psi_1|$, and since, as stated earlier, $\mathrm{tr}_a\left(|\psi_1\rangle\langle\psi_1|\right) = \mathrm{tr}_a\left(|\psi_2\rangle\langle\psi_2|\right)$ with $a \subset b$, it follows that $\mathrm{tr}_b\left(|\psi_1; \Delta t\rangle\langle\psi_1; \Delta t|\right) = \mathrm{tr}_b\left(|\psi_2\rangle\langle\psi_2|\right)$. Combining this with the previously mentioned relation $\mathrm{tr}_b\left(|\psi_1; \Delta t\rangle\langle\psi_1; \Delta t|\right) = \mathrm{tr}_b\left(|\psi_2; \Delta t\rangle\langle\psi_2; \Delta t|\right)$, we obtain $\mathrm{tr}_b\left(|\psi_2; \Delta t\rangle\langle\psi_2; \Delta t|\right) = \mathrm{tr}_b\left(|\psi_2\rangle\langle\psi_2|\right)$. This implies the existence of a unitary operator $\hat{V}_b$ supported in region $b$ such that $|\psi_2; \Delta t\rangle = e^{-i\hat{H}t}|\psi_2\rangle = \hat{V}_b|\psi_2\rangle$.

We have discussed three different meanings of "physical operations" above, each corresponding to a local unitary operator $\hat{U}_b$, $\hat{U}_b'$, and $\hat{V}_b$. However, it is worth noting that $|\psi_2; \Delta t\rangle$ and $|\psi_2\rangle$ include not only the system being operated on but also the apparatus (or the human) performing the operation. This may differ from interpretations in the previous literature, for example Refs. [37] and [35]. Nevertheless, it is natural, since isolating the apparatus (or the human) from $|\psi_2; \Delta t\rangle$ generally leaves the operated system no longer in a pure state, owing to the interaction between the apparatus (or the human) and the system. But for the sake of clarity and convenience in the following discussion, we will treat the apparatus as effectively independent of the quantum state $|\psi_2\rangle$, bearing in mind that this is an idealization. Based on the previous derivation, we may rephrase Witten's statement as: "If one asks not about mathematical operations in Hilbert space but about physical operations that are possible in the real world, then, since both the apparatus performing the operation and the system being operated upon are constituted by fundamental fields and are consequently subject to relativistic causality, the only physical way that one can modify a quantum state brings about a unitary transformation." Thus, even if we forcibly treat the apparatus as independent of the quantum state, causality directly leads us to the conclusion that a physical operation must be equivalent to a unitary operator supported in the region where the apparatus is located. In this manner, we circumvent the need for Witten's "perturbing the Hamiltonian" (which is in fact equivalent to the third hypothesis of Ref. [35] mentioned in Section 2.2), yet arrive at the same final conclusion that physical operations must admit a unitary representation.

# 4 Schrödinger's Cat Paradox and A Possible Interpretation of Quantum Mechanics

In Section 1, we have already emphasized the central role of Schrödinger's cat paradox in interpretations of quantum mechanics. In order to resolve this paradox, a wide variety of interpretations of quantum mechanics have been proposed. Among them, the many-worlds interpretation [33, 38] aligns perfectly with the completeness of quantum mechanics; however, it requires one to accept the bizarre "many-worlds scenario" that arises from Schrödinger's cat paradox. In this section, we argue that the traditional derivation of the Schrödinger's cat paradox becomes problematic when examined within the framework of quantum field theory. This suggests a possible pathway toward an interpretation that preserves the completeness of quantum theory while obviating the need for the "many-worlds scenario". For the sake of discussion, we provisionally refer to this potential framework as the "one-world interpretation". We also discuss how the one-world interpretation differs from traditional hidden-variable theories and analyze its compatibility with relativistic causality.

## 4.1 The traditional derivation of Schrödinger's cat paradox

Assuming that quantum mechanics is complete, the entire universe can be governed by quantum mechanics and described by a single quantum state, then even complicated measurement processes follow unitary time evolution. In a measurement experiment, we single out the microscopic system S to be measured and treat the remainder of the universe as the detector D. The detector D is manifestly macroscopic, and different detector readouts are macroscopically distinguishable. Let the initial quantum state of the detector D be $|d_0\rangle$, and the initial quantum state of the system S be $|s\rangle$. In quantum mechanics, there is a fundamental assumption often cited as an additional "axiom (0)" [39, 40]: "The states of composite quantum systems are represented by a vector in the tensor product of the Hilbert spaces of its components." Then, according to "axiom (0)", before the measurement begins, the composite system consisting of the system S and the detector D (i.e., the entire universe) can be written as $|s\rangle|d_0\rangle$.

After the measurement, different detector readouts are labeled by $d_i$, and the corresponding quantum state of the composite system is denoted by $|d_i\rangle$ with $i \neq 0$. Note that we do not care about the post-measurement state of the microscopic system; in fact, $|d_i\rangle$ already represents the quantum state of the entire universe. Let $|s_i\rangle$ be the state of the microscopic system S that leads deterministically to the measurement outcome $d_i$—that is, whenever the detector D measures the microscopic system S in $|s_i\rangle$ it yields the fixed readout $d_i$ and no other outcome. Thus, the measurement process can be expressed as

$$|s_i\rangle|d_0\rangle \rightarrow |d_i\rangle \ . \tag{11}$$

In general, the initial state of the system S is a superposition $|s\rangle = \sum_i c_i|s_i\rangle$. Then, under unitary time evolution during the measurement process, the initial state of the composite system

(i.e., the entire universe) evolves as

$$|s\rangle|d_0\rangle = \sum_i c_i|s_i\rangle|d_0\rangle \rightarrow \sum_i c_i|d_i\rangle \ . \tag{12}$$

The final state is a superposition of different measurement outcomes, unable to yield a unique classical measurement result. In the Schrödinger's cat experiment, such a state corresponds to the cat being both dead and alive simultaneously.

This shows that unitary evolution cannot turn possible results into actual results, which is in contradiction with real measurement experiments, where only a single outcome is ever observed. Various interpretations of quantum mechanics have been proposed to resolve this paradox [33], including the well-known interpretation proposed by Wigner, where collapse occurs whenever a conscious human being observes a detector in a superposed state. Here we would like to highlight in particular the many-worlds interpretation, which holds that each measurement outcome $|d_i\rangle$ displayed in Eq. (12) is physically real, with different measurement outcomes corresponding to different worlds. The greatest strength of the many-worlds interpretation is that it maintains the completeness of quantum mechanics, a standpoint that is fully consistent with the basic philosophy of this paper.

## 4.2 Schrödinger's cat paradox in quantum field theory

The above derivation of the cat paradox is based on non-relativistic quantum mechanics. However, in quantum field theory, the system $S$ and the detector $D$ are composed of the same fundamental fields. Therefore strictly speaking, we cannot fully distinguish between the system $S$ and the detector $D$. Even if we consider quantum fields at different spatial points as different field operators, the interactions between fields at adjacent points lead to strong entanglement between different points in the quantum state. Besides, the ubiquitous and constant interactions among fields cause field mixing, making it impossible to disentangle different types of fields [41]. Even the vacuum state, as fundamental and simple as it is, cannot be expressed as the direct product of "vacuum states" of different regions, nor can it be expressed as the direct product of "vacuum states" of different types of fields (bare vacuum). Therefore, in the framework of quantum field theory, the initial state of the "composite" system formed by the detector $D$ and the system being measured $S$ cannot be written in the form of $|s\rangle|d_0\rangle$.

Although we cannot strictly distinguish the detector $D$ from the system being measured $S$ at the level of quantum states, we can distinguish them in space using reduced density matrices. The reduced density matrices characterize the information in a particular region. When the system $S$ and the detector $D$ do not overlap spatially, we can describe their respective states using reduced density matrices. Let the quantum state $|s_1, d_0\rangle$ represent the system $S$ in state $s_1$ and the detector $D$ in state $d_0$, and let $|s_2, d_0\rangle$ represent the system $S$ in state $s_2$ and the detector $D$ in state $d_0$. Suppose the detector $D$ is located in region $A$ while the system $S$ is in region $a$. Since the detectors of both quantum states $|s_1, d_0\rangle$ and $|s_2, d_0\rangle$ are in state $d_0$, we have

$$\text{tr}_a\left(|s_1, d_0\rangle\langle s_1, d_0|\right) = \text{tr}_a\left(|s_2, d_0\rangle\langle s_2, d_0|\right) \ . \tag{13}$$

According to the derivation in Section 2.2, it is known that there exists a unitary operator $\hat{U}_a$ supported in region $a$, such that $|s_2, d_0\rangle = \hat{U}_a|s_1, d_0\rangle$.

Next, we consider a quantum state $|\psi_3\rangle \equiv c_1|s_1, d_0\rangle + c_2|s_2, d_0\rangle$. Under time evolution, this state would evolve into a superposition of different measurement outcomes, i.e., the Schrödinger-cat state $c_1|d_1\rangle + c_2|d_2\rangle$. We will argue, however, that due to the constraints imposed by causality on physical operations, it may not be possible to prepare the quantum state $|\psi_3\rangle$ in a generic measurement experiment.

To better contrast the differences between traditional non-relativistic quantum mechanics and quantum field theory, let us first assume that $|s_1, d_0\rangle$ and $|s_2, d_0\rangle$ can be written as direct-product states, namely $|s_1\rangle|d_0\rangle$ and $|s_2\rangle|d_0\rangle$. In this case, $|\psi_3\rangle = \left(c_1|s_1\rangle + c_2|s_2\rangle\right)|d_0\rangle$. It is evident that $\mathrm{tr}_a\left(|\psi_3\rangle\langle\psi_3|\right) = \mathrm{tr}_a\left(|s_1, d_0\rangle\langle s_1, d_0|\right)$ and thus there exists a unitary operator $\hat{V}_a$ supported in region $a$ such that $|\psi_3\rangle = \hat{V}_a|s_1, d_0\rangle$. According to Section 3, the unitary operator $\hat{V}_a$ supported in region $a$ may be interpreted as representing a physical operation localized in region $a$. Therefore, by manipulating the microscopic system located in region $a$, one can prepare the state $|\psi_3\rangle$ and thereby realize a Schrödinger'cat state.

However, when $|s_1, d_0\rangle$ and $|s_2, d_0\rangle$ cannot be written in direct-product form, in general there does not exist a unitary operator $\hat{V}_a$ supported in region $a$ such that $|\psi_3\rangle = \hat{V}_a|s_1, d_0\rangle$. This shows that, due to the constraints imposed by causality on physical operations, it is impossible to obtain $|\psi_3\rangle$ solely by manipulating the measured microscopic system $S$ localized in region $a$. To illustrate the role played by entanglement in this context, we present a simple quantum-mechanical example. Consider two systems, $a$ and $A$. The basis vectors describing system $a$ are $|\varphi_1\rangle_a, |\varphi_2\rangle_a, |\varphi_3\rangle_a$, and the basis vectors describing system $A$ are $|\phi_1\rangle_A, |\phi_2\rangle_A, |\phi_3\rangle_A$. We consider the following quantum state of the composite system consisting of $a$ and $A$:

$$|s_1, d_0\rangle = b_1|\varphi_1\rangle_a|\phi_1\rangle_A + b_2|\varphi_2\rangle_a|\phi_2\rangle_A + b_3|\varphi_3\rangle_a|\phi_3\rangle_A \ . \tag{14}$$

The statement that system $A$ is in state $d_0$ means that system $A$ is described by the reduced density matrix

$$\rho_A = \mathrm{tr}_a\left(|s_1, d_0\rangle\langle s_1, d_0|\right) = b_1^* b_1|\phi_1\rangle_A {}_A\langle\phi_1| + b_2^* b_2|\phi_2\rangle_A {}_A\langle\phi_2| + b_3^* b_3|\phi_3\rangle_A {}_A\langle\phi_3| \ . \tag{15}$$

Similarly, the statement that system $a$ is in state $s_1$ means that system $a$ is described by the reduced density matrix $\rho_a = b_1^* b_1|\varphi_1\rangle_a {}_a\langle\varphi_1| + b_2^* b_2|\varphi_2\rangle_a {}_a\langle\varphi_2| + b_3^* b_3|\varphi_3\rangle_a {}_a\langle\varphi_3|$. We define a unitary operator $\hat{U}_a$ acting on system $a$ such that

$$\hat{U}_a|\varphi_1\rangle_a = |\varphi_2\rangle_a, \qquad \hat{U}_a|\varphi_2\rangle_a = |\varphi_3\rangle_a, \qquad \hat{U}_a|\varphi_3\rangle_a = |\varphi_1\rangle_a \ .$$

We then define $|s_2, d_0\rangle$ as

$$|s_2, d_0\rangle = \hat{U}_a|s_1, d_0\rangle = b_1|\varphi_2\rangle_a|\phi_1\rangle_A + b_2|\varphi_3\rangle_a|\phi_2\rangle_A + b_3|\varphi_1\rangle_a|\phi_3\rangle_A \ . \tag{16}$$

Thus, the statement that system $a$ is in state $s_2$ means that it is described by the reduced density matrix $\rho_a' = b_3^* b_3|\varphi_1\rangle_a {}_a\langle\varphi_1| + b_1^* b_1|\varphi_2\rangle_a {}_a\langle\varphi_2| + b_2^* b_2|\varphi_3\rangle_a {}_a\langle\varphi_3|$. Moreover, since the unitary operator $\hat{U}_a$ acts only on system $a$, the reduced density matrix of system $A$ remains unchanged, i.e.

$$\rho_A' = \rho_A \quad \text{or equivalently} \quad \mathrm{tr}_a\left(|s_1, d_0\rangle\langle s_1, d_0|\right) = \mathrm{tr}_a\left(|s_2, d_0\rangle\langle s_2, d_0|\right) \ .$$

This shows that system $A$ corresponding to $|s_2, d_0\rangle$ indeed remains in state $d_0$. Notice that an interesting feature of this example is that $|s_1, d_0\rangle$ and $|s_2, d_0\rangle$ are orthogonal, i.e. $\langle s_1, d_0|s_2, d_0\rangle = 0$. Combining the definitions of $|s_1, d_0\rangle$ and $|s_2, d_0\rangle$, we can write their superposition as

$$
\begin{aligned}
|\psi_3\rangle &= c_1|s_1, d_0\rangle + c_2|s_2, d_0\rangle \\
&= |\varphi_1\rangle_a \Big(c_1 b_1 |\phi_1\rangle_A + c_2 b_3 |\phi_3\rangle_A\Big) + |\varphi_2\rangle_a \Big(c_1 b_2 |\phi_2\rangle_A + c_2 b_1 |\phi_1\rangle_A\Big) + |\varphi_3\rangle_a \Big(c_1 b_3 |\phi_3\rangle_A + c_2 b_2 |\phi_2\rangle_A\Big) .
\end{aligned}
\tag{17}
$$

The reduced density matrix of system $A$ corresponding to $|\psi_3\rangle$ is

$$
\begin{aligned}
\rho_A'' = {}& b_1^* b_1 |\phi_1\rangle_A {}_A\langle\phi_1| + b_2^* b_2 |\phi_2\rangle_A {}_A\langle\phi_2| + b_3^* b_3 |\phi_3\rangle_A {}_A\langle\phi_3| \\
&+ c_2^* c_1 b_3^* b_1 |\phi_1\rangle_A {}_A\langle\phi_3| + c_1^* c_2 b_1^* b_3 |\phi_3\rangle_A {}_A\langle\phi_1| + c_1^* c_2 b_2^* b_1 |\phi_1\rangle_A {}_A\langle\phi_2| \\
&+ c_2^* c_1 b_1^* b_2 |\phi_2\rangle_A {}_A\langle\phi_1| + c_2^* c_1 b_2^* b_3 |\phi_3\rangle_A {}_A\langle\phi_2| + c_1^* c_2 b_3^* b_2 |\phi_2\rangle_A {}_A\langle\phi_3| .
\end{aligned}
\tag{18}
$$

Evidently, $\rho_A'' \neq \rho_A$, i.e. $\mathrm{tr}_a\left(|\psi_3\rangle\langle\psi_3|\right) \neq \mathrm{tr}_a\left(|s_1, d_0\rangle\langle s_1, d_0|\right)$, which shows that there does not exist a unitary operator $\hat{V}_a$ acting only on system $a$ such that $|\psi_3\rangle = \hat{V}_a|s_1, d_0\rangle$. This also demonstrates that it is impossible to obtain a Schrödinger-cat state solely by manipulating the measured microscopic system $S$ localized in region $a$.

We can also give a fully quantum field–theoretic example to illustrate the differences between quantum field theory and quantum mechanics in the context of the Schrödinger's cat problem. We consider a special case in which $d_0$ in $|s_1, d_0\rangle$ and $|s_2, d_0\rangle$ is taken to be the vacuum, i.e.,

$$
\mathrm{tr}_a\left(|s_1, d_0\rangle\langle s_1, d_0|\right) = \mathrm{tr}_a\left(|s_2, d_0\rangle\langle s_2, d_0|\right) = \mathrm{tr}_a\left(|\Omega\rangle\langle\Omega|\right) ,
$$

meaning that the entire space contains only the measured microscopic system $S$ but no macroscopic detector $D$. Thus, we can directly abbreviate $|s_1, d_0\rangle$ and $|s_2, d_0\rangle$ as $|s_1\rangle$ and $|s_2\rangle$. Since $|s_1\rangle$ and $|s_2\rangle$ describe microscopic systems, in nonrelativistic quantum mechanics we usually assume that we can prepare the superposition state $|s_1\rangle + |s_2\rangle$. However, in quantum field theory our operations must be represented by unitary local operators, and therefore even if $s$ is a microscopic system, we are not necessarily able to prepare the state $|s_1\rangle + |s_2\rangle$. For the sake of concrete calculations, we take $|s_1\rangle$ and $|s_2\rangle$ to be coherent states in the free real scalar field theory.

We first briefly introduce some basic properties of coherent states, which have been proven in Ref. [42]. Let $|\phi\rangle$ be an eigenstate of the field operator $\hat{\phi}(\boldsymbol{x})$ with eigenvalue $\phi(\boldsymbol{x})$. The coherent state $|\phi_{\mathrm{class}}, \pi_{\mathrm{class}}\rangle$ in the representation expanded by $|\phi\rangle$ is expressed as

$$
\begin{aligned}
\langle\phi|\phi_{\mathrm{class}}, \pi_{\mathrm{class}}\rangle = {}& \mathcal{N}' \exp\left\{-\frac{1}{2}\int d^3x d^3y \, \mathcal{E}(\boldsymbol{x}, \boldsymbol{y})\left[\phi(\boldsymbol{x}) - \phi_{\mathrm{class}}(\boldsymbol{x})\right]\left[\phi(\boldsymbol{y}) - \phi_{\mathrm{class}}(\boldsymbol{y})\right]\right\} \\
&\times \exp\left\{i\int d^3x \, \pi_{\mathrm{class}}(\boldsymbol{x})\phi(\boldsymbol{x})\right\} ,
\end{aligned}
\tag{19}
$$

where $\mathcal{N}'$ is the normalization coefficient, and $\mathcal{E}(\boldsymbol{x}, \boldsymbol{y})$ is defined by

$$
\mathcal{E}(\boldsymbol{x}, \boldsymbol{y}) \equiv \int \frac{d^3p}{(2\pi)^3} e^{i\boldsymbol{p}\cdot(\boldsymbol{x}-\boldsymbol{y})} E_{\boldsymbol{p}} , \qquad E_{\boldsymbol{p}} = \sqrt{\boldsymbol{p}^2 + m^2} .
\tag{20}
$$

The coherent state $|\phi_{\text{class}}, \pi_{\text{class}}\rangle$ is determined by the corresponding classical fields $\phi_{\text{class}}$ and $\pi_{\text{class}}$ appearing in Eq. (19). Dividing the space into two regions, denoted as region $A$ and region $a$, and tracing out region $a$, we can obtain the reduced density matrix on region $A$ as

$$
\begin{aligned}
\int \mathcal{D}\phi_a \; \rho(\phi, \phi') \Big|_{\phi(\boldsymbol{x}_a)=\phi'(\boldsymbol{x}_a),\, \boldsymbol{x}_a \in a} &= \int \mathcal{D}\phi_a \; \langle \phi | \phi_{\text{class}}, \pi_{\text{class}} \rangle \langle \phi_{\text{class}}, \pi_{\text{class}} | \phi' \rangle \Big|_{\phi(\boldsymbol{x}_a)=\phi'(\boldsymbol{x}_a),\, \boldsymbol{x}_a \in a} \\
&= |\mathcal{N}'|^2 \exp \left\{ i \int_A d^3x \; \pi_{\text{class}}(\boldsymbol{x}_A) \Big[ \phi(\boldsymbol{x}_A) - \phi'(\boldsymbol{x}_A) \Big] \right\} \\
&\quad \times \exp \left\{ -\frac{1}{2} \int_A d^3x \int_A d^3y \; \mathcal{E}(\boldsymbol{x}_A, \boldsymbol{y}_A) \left[ \phi(\boldsymbol{x}_A) - \phi_{\text{class}}(\boldsymbol{x}_A) \right] \left[ \phi(\boldsymbol{y}_A) - \phi_{\text{class}}(\boldsymbol{y}_A) \right] \right\} \\
&\quad \times \exp \left\{ -\frac{1}{2} \int_A d^3x \int_A d^3y \; \mathcal{E}(\boldsymbol{x}_A, \boldsymbol{y}_A) \left[ \phi'(\boldsymbol{x}_A) - \phi_{\text{class}}(\boldsymbol{x}_A) \right] \left[ \phi'(\boldsymbol{y}_A) - \phi_{\text{class}}(\boldsymbol{y}_A) \right] \right\} \\
&\quad \times \int \mathcal{D}\varphi_a \; \exp \left\{ - \int_a d^3x \int_a d^3y \; \mathcal{E}(\boldsymbol{x}_a, \boldsymbol{y}_a) \varphi(\boldsymbol{x}_a) \varphi(\boldsymbol{y}_a) \right\} \\
&\quad \times \exp \left\{ - \int_a d^3x \int_A d^3y \; \mathcal{E}(\boldsymbol{x}_a, \boldsymbol{y}_A) \varphi(\boldsymbol{x}_a) \left[ \phi(\boldsymbol{y}_A) - \phi_{\text{class}}(\boldsymbol{y}_A) \right] \right\} \\
&\quad \times \exp \left\{ - \int_a d^3x \int_A d^3y \; \mathcal{E}(\boldsymbol{x}_a, \boldsymbol{y}_A) \varphi(\boldsymbol{x}_a) \left[ \phi'(\boldsymbol{y}_A) - \phi_{\text{class}}(\boldsymbol{y}_A) \right] \right\} .
\end{aligned}
\tag{21}
$$

where $\boldsymbol{x}_A \in A$ and $\boldsymbol{x}_a \in a$, and $\int \mathcal{D}\phi_a$ denotes the functional integration over the field configurations restricted to region $a$ (i.e., over $\phi(\boldsymbol{x}_a)$ for all $\boldsymbol{x}_a \in a$). Note that in Eq. (21), there are no appearances of $\phi_{\text{class}}(\boldsymbol{x}_a)$ and $\pi_{\text{class}}(\boldsymbol{x}_a)$. This indicates that Eq. (21) only depends on the classical fields $\phi_{\text{class}}(\boldsymbol{x}_A)$ and $\pi_{\text{class}}(\boldsymbol{x}_A)$ in region $A$. Therefore, the reduced density matrix of a coherent state in a given region is completely determined by the classical fields in that region. In other words, if two classical fields are identical in a certain region, then the reduced density matrices of their corresponding coherent states are also identical in that region.

We take $|s_1\rangle = |\phi_{\text{class}}, \pi_{\text{class}}\rangle$ and $|s_2\rangle = |\phi'_{\text{class}}, \pi'_{\text{class}}\rangle$, where the classical fields satisfy $\phi_{\text{class}}(\boldsymbol{x}_A) = \pi_{\text{class}}(\boldsymbol{x}_A) = \phi'_{\text{class}}(\boldsymbol{x}_A) = \pi'_{\text{class}}(\boldsymbol{x}_A) = 0$ for $\boldsymbol{x}_A \in A$. Then, according to the properties of coherent states introduced above (noting that the vacuum state $|\Omega\rangle$ is a special coherent state with $\phi_{\text{class}} = \pi_{\text{class}} = 0$), we have

$$
\text{tr}_a \left( |s_1\rangle \langle s_1| \right) = \text{tr}_a \left( |s_2\rangle \langle s_2| \right) = \text{tr}_a \left( |\Omega\rangle \langle \Omega| \right)
$$

This suggests that we can obtain $|s_1\rangle$ and $|s_2\rangle$ from the vacuum through unitary operators supported in region $a$. At the same time, it implies that there exists a unitary operator $\hat{U}_a$ supported in region $a$ such that $|s_2\rangle = \hat{U}_a |s_1\rangle$. Let $\psi_3 = |s_1\rangle + |s_2\rangle$. Since both $|s_1\rangle$ and $|s_2\rangle$ are coherent states, the reduced density matrix of $\psi_3$ can be directly written down. A straightforward calculation [42] shows that

$$
\text{tr}_a \left( |\psi_3\rangle \langle \psi_3| \right) \neq \text{tr}_a \left( |\Omega\rangle \langle \Omega| \right) = \text{tr}_a \left( |s_1\rangle \langle s_1| \right)
$$

This indicates that there does not exist a unitary operator $\hat{V}_a$ supported in region $a$ such that $|\psi_3\rangle = \hat{V}_a |s_1\rangle$. This shows that even if $S$ is a microscopic system localized in region $a$ and we are able to prepare its two states $|s_1\rangle$ and $|s_2\rangle$, we still cannot obtain their superposition $\psi_3 = |s_1\rangle + |s_2\rangle$

by physical operations confined to region $a$ due to the entanglement between regions $a$ and $A$. This is markedly different from conventional nonrelativistic quantum mechanics.

Consequently, in quantum field theory, we have $|s, d_0\rangle \neq \sum_i c_i |s_i, d_0\rangle$. As a result, even if we can write (11) as $|s_i, d_0\rangle \rightarrow |d_i\rangle$, we cannot derive a formula similar to (12) (i.e., $|s, d_0\rangle = \sum_i c_i |s_i, d_0\rangle \rightarrow \sum_i c_i |d_i\rangle$). Instead, we have

$$|s, d_0\rangle \nrightarrow \sum_i c_i |d_i\rangle \ . \tag{22}$$

This indicates that Schrödinger's cat paradox cannot be derived in quantum field theory in the same way as in nonrelativistic quantum mechanics.

The above discussion actually shows that the ability to express the initial state in a product form is crucial for deriving the Schrödinger's cat paradox. The traditional derivation of the Schrödinger's cat paradox relies on two measurement-related ingredients: (1) the measurement amplifies microscopic differences into macroscopic classical differences, i.e., $|s_i, d_0\rangle \rightarrow |d_i\rangle$; (2) one can freely manipulate the microscopic system being measured, namely by applying any unitary operator $\hat{V}_a$ supported in region $a$. An initial state in direct product form allows these two ingredients to be combined so that, one can obtain a superposition of vastly different macroscopic states by acting only on the microscopic system, thereby leading to the paradox. In fact, in quantum field theory there also exist states of regions $a$ and $A$ that are not entangled (for example, eigenstates of field operators). For such states, one can, in principle, use unitary operations supported in region $a$ to derive the Schrödinger's cat paradox (if the amplification effect still exists). However, such non-entangled states are extremely nonclassical and idealized to the extent that they cannot naturally exist in the real world — even the vacuum state, the most fundamental state, is itself a highly entangled state.

It is worth noting that we do not deny the existence of superposition states of vastly different macroscopic states — for example, a quantum superposed cat that is both dead and alive at the same time. We only contend that such a quantum superposed cat cannot be produced through a simple, traditional measurement process. More specifically, although a state like $|\psi_3\rangle$, which could give rise to a paradox, may exist, it does not occur in actual measurement experiments. We do not deny that it might be possible to create such special states as $|\psi_3\rangle$ by using very complex and delicate nonlocal operations; however, that would constitute an entirely new type of experiment and would not contradict current simple measurement experiments.

## 4.3   The preparation apparatus and the Schrödinger's cat paradox

The discussion above neglects the apparatus that prepares (or manipulates) the microscopic system $S$. We did not explicitly include the preparation apparatus in the main argument because the traditional derivation of Schrödinger's cat paradox also does not take the preparation apparatus into account. In fact, since the preparation apparatus is manifestly macroscopic, its inclusion further elucidates why the Schrödinger's cat paradox may not arise.

Suppose the preparation apparatus has three buttons: button 1 prepares the microscopic system in state $s_1$ (so after measurement the detector will be in state $d_1$); button 2 prepares $s_2$

(so after measurement the detector will be in state $d_2$); and button 3 prepares the superposition $s_1 + s_2$ (so after measurement the detector would be either $d_1$ or $d_2$). If the initial state of the preparation apparatus itself were a superposition of "button 1 pressed" and "button 2 pressed", then after measurement the detector would indeed end up in a superposition $d_1 + d_2$ (i.e. the alive-cat + dead-cat state). However, in that case the initial state of the preparation apparatus would already be a macroscopic superposition. This does not contradict actual experimental practice, in which the initial state of the preparation device is not a macroscopic superposition. In real experiments, one presses button 3 to prepare a superposition of microscopic states. There is then no reason to expect that pressing button 3 should result in the detector being in a superposition of $d_1$ and $d_2$. Hence no paradox arises.

The above discussion reflects, to some extent, the core message of our paper: the quantum superposition of an experiment preparing the microscopic system in state $s_1$ and an experiment preparing it in state $s_2$, is not equivalent to an experiment that prepares the microscopic system in the superposed state $s_1 + s_2$.

## 4.4   One-world interpretation

To clarify the discussion, we need to provide a more detailed explanation of "macroscopic state". A macroscopic state refers to a state described using classical concepts, such as a dead cat, a live cat, classical fields, a detector displaying measurement outcomes, and so on. Since classical descriptions are generally more ambiguous compared to quantum descriptions, a macroscopic state often corresponds to many quantum states. For example, many quantum states can describe "a dead cat". Even detectors displaying measurement outcomes correspond to multiple quantum states. This is because quantum states encompass not only specific displayed measurement readings but also factors like the average temperature of the detector, ubiquitous phonons in the detector, and so forth. More precisely, quantum states include the states of each atomic and molecular constituent composing the macroscopic detector. Of course, there are also instances where each "macroscopic state" corresponds to a single quantum state. For example, in free scalar field theory, the classical vacuum ($\phi = \pi = 0$) corresponds uniquely to the vacuum state $|\Omega\rangle$ in quantum field theory.

As demonstrated in the previous subsection, there may be no Schrödinger's cat paradox in quantum field theory. Therefore, a crucial possibility arises: After the measurement process, the entire composite system evolves into a definite macroscopic state, rather than a superposition of macroscopic states. In other words, in the Schrödinger's cat experiment, the cat would theoretically evolve into either being alive or being dead, rather than into a superposition of being both alive and dead. We don't need (consciousness) to observe a detector to collapse it into a specific macroscopic state; instead, the detector automatically evolves to a definite macroscopic state. This naturally leads to a new interpretation of quantum mechanics: quantum mechanics is complete, and the time evolution of quantum states can describe measurement processes, yielding a unique measurement outcome for each measurement. This new interpretation and the many-worlds interpretation both consider quantum mechanics to be complete [33]. However, in contrast to the many-worlds interpretation, this new interpretation argue that there aren't "many worlds" but rather there is only "one world". Therefore, we may figuratively refer to this new

interpretation as the "one-world interpretation". It is worth noting that this interpretation does not oppose the existence of superposition states of macroscopic states that are vastly different, such as a quantum superposed cat that is dead and alive at the same time. However, this interpretation suggests that it is not possible to prepare a quantum superposed cat that is dead and alive by following the steps of the Schrödinger's cat experiment.

A natural follow-up question arises: where does the probability in quantum mechanics come from? Looking back, the reason why probability was used to describe experimental results in the early days of quantum mechanics was that experiments with the same "initial setup" yielded different results. Only by statistically analyzing the results of multiple experiments could patterns emerge. In fact, however, the initial quantum state of each experiment is different. The "initial setup" includes the quantum state of the system being measured and the macroscopic state of the detector, but a macroscopic state can correspond to multiple quantum states. Because we cannot control that every atom and molecule in the macroscopic detector remains unchanged, the initial quantum state of the entire system is not the same in each repeated experiment, which leads to the randomness of measurement results. Ref. [43] formulates a mechanism for how the first droplet in a cloud chamber track arises, which provides a concrete example illustrating that the randomness of detection outcomes originates from the initial conditions that include the quantum state of the detector itself. The physics of a Geiger counter and a cloud chamber have a lot in common, then similar analysis can be applied to a Geiger counter as well [44].

If we toss a coin multiple times and statistically analyze the results, we can obtain a stable probability distribution, from which we can extract some information about the coin. For example, if the probability of landing heads up is not $1/2$, it indicates that the coin may be asymmetric. In quantum mechanics, measurements follow a similar principle. Through multiple repetitions of experiments and statistical analysis of the results, the randomness of the initial state can be averaged out, revealing information that remains constant across each experiment. Since the state of the system being measured remains the same in each experiment, the randomness inherent in the detector itself is averaged out, thereby revealing information about the system being measured.

Regarding the measurement process, a more specific but qualitative description is as follows. When the system being measured interacts with the detector, the composite system is in a highly unstable state. Subsequently, the entire system rapidly evolves into a stable macroscopic state, namely the state where the detector displays a specific reading. If the system being measured is in an eigenstate of the observable before the measurement, then the stable macroscopic state of the composite system formed by the system and the detector is unique, and the composite system naturally evolves to a definite outcome. If the system being measured is in a superposition of eigenstates of the observable before the measurement, then there are multiple stable state of the composite system. Because the composite system is in a highly unstable state during the measurement process, even tiny changes in the initial state lead to the composite system evolving to different stable states, leading to various possible measurement outcomes. Therefore, the determination of which outcome the composite system evolves to depends on the precise initial quantum state of the composite system.

The mechanism for the formation of the first droplet in the Cloud Chamber, as presented in Ref. [43], serves as an example illustrating the measurement process described above. Specifically, when the decay product appear in the Cloud Chamber, the composite system is in a highly

unstable state. At some time and in some location, a cluster of fortuitous size appears (as a result of thermal fluctuations) with an enormous ionization cross section, then the decay product wavefunction quickly becomes collimated at the location of the cluster, and the system rapidly evolves into a stable state—the state where the first droplet forms.

Finally, it is worth mentioning that in the one-world interpretation, there is no wave function collapse. Wave function collapse arises from treating the system under measurement as an independent entity from the detector at the quantum level, and describing the system being measured with a separate quantum state (wave function). For instance, in (12), if we describe the system being measured as $|s\rangle$ and it evolves into $|s_i\rangle$ after the measurement process, we would say that the system has collapsed from $|s\rangle$ to $|s_i\rangle$. However, in quantum field theory, since all matter in the world is composed of the same fundamental fields and there are interactions between the detector and the system being measured, the fields that make up the system being measured must also be part of the detector's composition according to the field mixing effect [41]. Therefore, the system being measured cannot be described by a separate quantum state as in the non-relativistic case, especially when the measured system and the detector overlap in space and merge into a single entity during their interaction. Consequently, there is no Traditional concept of wave function collapse in the one-world interpretation.

## 4.5 Reeh–Schlieder theorem and macroscopic definiteness

As pointed out in Section 4.4, the randomness of measurement outcomes originates from the randomness of the initial state of the entire system. In other words, it is impossible for us to control the state of every atom and molecule in the detector to be exactly the same in each experimental run, and this lack of microscopic control is what gives rise to the apparent randomness of measurement results. This observation also implies that we should distinguish the states of a macroscopic detector more carefully. For example, the detector's initial state $d_0$ discussed in Section 4.2 is actually a coarse-grained macroscopic state: many different microscopic states correspond to the same macroscopic description (just as many different quantum states of an isolated gas can correspond to the same thermodynamic macrostate). We denote these microscopic states by $d_0^{(1)}, d_0^{(2)}, d_0^{(3)}, ....$ The corresponding final states of the entire system are then $d_i^{(1)}, d_i^{(2)}, d_i^{(3)}, ....$ More concretely, the measurement process can be written as

$$|s_i, d_0^{(k)}\rangle \to |d_i^{(k)}\rangle . \tag{23}$$

Section 4.2 argues that the initial state of a measurement experiment cannot, in general, be written as a simple superposition $|s, d_0^{(1)}\rangle \neq |s_1, d_0^{(1)}\rangle + |s_1, d_0^{(1)}\rangle$ and therefore one cannot conclude that $|s, d_0^{(1)}\rangle$ will evolve into a Schrödinger's cat state (i.e., $|d_1^{(1)}\rangle + |d_2^{(1)}\rangle$) after the measurement. However, a more general possibility is quite plausible: the initial state of the measurement experiment might take the form

$$|s, d_0^{(1)}\rangle = \sum_n a_n |s_1, d_0^{(n)}\rangle + \sum_k b_k |s_2, d_0^{(k)}\rangle . \tag{24}$$

Then, according to Eq. (23), the measurement process with the initial state $|s, d_0^{(1)}\rangle$ can be written

as

$$|s, d_0^{(1)}\rangle \rightarrow \sum_n a_n |d_1^{(n)}\rangle + \sum_k b_k |d_2^{(k)}\rangle \ . \tag{25}$$

From Eq. (25) we see that the final state of the measurement process is a superposition of macroscopically distinct classical configurations. However, unlike the usual Schrödinger's cat state, this is no longer merely a superposition of two quantum states, but rather a superposition of many quantum states. Does this imply that quantum mechanics itself cannot provide a unique measurement outcome?

We argue that, within quantum field theory, the final states appearing in Eq. (25) may themselves belong to certain classes of classical macrostates — i.e., states that correspond to definite macroscopic outcomes. To illustrate this point we consider coherent states in a free field theory. Although writing down the quantum state corresponding to a generic macroscopic classical configuration is difficult, it is well known that coherent states are the unambiguous quantum counterparts of classical field configurations. In particular, from Eq. (19) one can obtain the expression for the coherent state $|\phi_{\text{class}}, \pi_{\text{class}}\rangle$ corresponding to the classical fields $\phi_{\text{class}}$ and $\pi_{\text{class}}$.

We choose two disjoint compact regions, denoted by $a$ and $b$, which are separated by a large distance. We consider two classical field configurations. One is a classical field configuration that is nonvanishing only in region $a$ (i.e., $\pi(x) = \phi(x) = 0$ for any $x \notin a$); we denote this configuration by $d_1$. The other is a classical field configuration that is nonvanishing only in region $b$; we denote this configuration by $d_2$. Since regions $a$ and $b$ do not overlap and are far apart, the localized classical field configurations $d_1$ and $d_2$ are clearly distinguishable.

According to Eq. (21), a classical field configuration localized in a given region corresponds to a coherent state that is also localized in that region. Therefore, all quantum states $|d_1^{(n)}\rangle$ corresponding to the classical configuration $d_1$ are localized in region $a$, while all quantum states $|d_2^{(k)}\rangle$ corresponding to the classical configuration $d_2$ are localized in region $b$. Evidently, $|d_1^{(n)}\rangle + |d_2^{(k)}\rangle$ is a Schrödinger's cat state — the system is simultaneously localized in region $a$ and region $b$. The question now is: what about the more general superposition $\sum_n a_n |d_1^{(n)}\rangle + \sum_k b_k |d_2^{(k)}\rangle$ ?

The Reeh–Schlieder theorem is a remarkable result in quantum field theory concerning the superposition principle. Here we adopt the formulation given in Ref. [45]. Roughly speaking, the Reeh–Schlieder theorem states that, given any region, any quantum state of a free field theory can be approximated arbitrarily well by superpositions of coherent states localized in that region. According to the Reeh–Schlieder theorem, the coherent states $|d_1^{(n)}\rangle$ localized in region $a$ can be superposed to generate any other quantum state. Consequently, hence we can adjust the coefficients $a_n$ such that:

$$\sum_n a_n |d_1^{(n)}\rangle = |d_1^{(n')}\rangle - \sum_k b_k |d_2^{(k)}\rangle \ , \tag{26}$$

where we require that the coefficients $b_k$ are not all zero and that the right-hand side does not vanish. Clearly, the coefficients $a_n$ are also not all zero. Rewrite Eq. (26) in the following form:

$$\sum_n a_n |d_1^{(n)}\rangle + \sum_k b_k |d_2^{(k)}\rangle = |d_1^{(n')}\rangle \ , \tag{27}$$

where not all the $b_k$ vanish, and not all the $a_n$ vanish either. Note that $|d_1^{(n')}\rangle$ denotes a coherent state localized in region $a$, corresponding to a classical field configuration that is nonzero only on region $a$ and vanishes on region $b$. Analogously, by adjusting the coefficients $b_k$ and $a_n$ we can replace the $|d_1^{(n')}\rangle$ appearing in Eq. (27) with some $|d_2^{(k')}\rangle$. This shows that even though the final state appearing in Eq. (25) contains superposition components associated with both the $d_1$ and $d_2$ states, the measurement outcome itself may still be a definite macroscopic state—either $d_1$ or $d_2$—rather than a Schrödinger's cat–like superposition.

This further indicates that, because the states in quantum field theory and their counterintuitive superposition properties differ markedly from those of ordinary quantum mechanics, studying the classical–quantum correspondence and analyzing the Schrödinger's cat paradox within the framework of quantum field theory may lead to different conclusions. Perhaps we truly do not need the many-worlds scenario in order for quantum mechanics to be a complete theory.

## 4.6 Violation of statistical independence and compatibility with relativistic causality

As demonstrated in Section 4.4, it can be seen that the one-world interpretation is actually a hidden-variable theory, where the hidden variables are the initial quantum states of the entire systems. This naturally raises the question: Will this interpretation be ruled out by experiments [46] related to Bell inequalities? Recalling the key formula used by Bell in deriving Bell's inequalities [47]:

$$P(\boldsymbol{a}, \boldsymbol{b}) = \int \mathrm{d}\lambda \, \rho(\lambda) A(\boldsymbol{a}, \lambda) B(\boldsymbol{b}, \lambda) \,, \tag{28}$$

where the probability distribution $\rho(\lambda)$ of the hidden variable $\lambda$ is independent of the macroscopic states of the detectors, i.e. $\boldsymbol{a}$ and $\boldsymbol{b}$ (specifically $\boldsymbol{a}$ and $\boldsymbol{b}$ are polarizer settings). In the one-world interpretation, the hidden variables $\lambda$ might superficially seem to correspond to the initial states of the detectors, however due to the failure of axiom (0), we cannot consider the two detectors separately, nor can we separate the detectors from the system being measured. Therefore, the hidden variables are actually the initial quantum state of the entire system $|\psi\rangle$. However, the quantum state $|\psi\rangle$ needs to satisfy certain constraints. In addition to ensuring that the initial states of the system being measured are consistent (i.e., their reduced density matrices of the system being measured are the same), it is also required that the corresponding macroscopic states of the detectors be $\boldsymbol{a}$ and $\boldsymbol{b}$, respectively. This constraint indicates that the range of values of the hidden variable $\lambda$ is not the same for different macroscopic states $\boldsymbol{a}$ and $\boldsymbol{b}$ of the detectors. Consequently, the distribution function $\rho(\lambda)$ is actually dependent on the macroscopic states $\boldsymbol{a}$ and $\boldsymbol{b}$, and should be denoted as $\rho(\lambda|\boldsymbol{a}, \boldsymbol{b})$ (More specifically, $\rho(\lambda|\boldsymbol{a}, \boldsymbol{b}) = \rho(|\psi\rangle|$the polarizer settings of $|\psi\rangle$ are $\boldsymbol{a}$ and $\boldsymbol{b}) \neq \rho(\lambda)$), which differs from Bell's assumption. In fact, the assumption that the hidden variables do not in any way depend on measurement settings, i.e. $\rho(\lambda|\boldsymbol{a}, \boldsymbol{b}) = \rho(\lambda)$, is commonly known as "Statistical Independence" [48,49]. Therefore, although the one-world interpretation is a hidden variable theory, it violate Statistical Independence and does not lead to Bell's inequalities as in Ref. [47], and as a result, it will not be ruled out by experiments related to Bell's inequality.

In addition, the one-world interpretation does not belong to the category of "nonlocal hidden-variable theory" as defined by Leggett in Ref. [50], because the nonlocal hidden-variable theory defined by Leggett satisfies Statistical Independence. Therefore, the one-world interpretation is not excluded by experiments [51] violating the inequality proposed by Leggett [50].

In Section 3, it was proved that a physical operation must be equivalent to a local unitary operator. In the Sorkin-type impossible measurements problem, such a unitary operation is also referred to as a "kick". Studying the effect of a kick on measurements that are spacelike separated is central to the Fermi two-atom problem and the Sorkin-type impossible measurements problem. In the Fermi two-atom problem, there is only one measurement, and there are no other measurements between the kick and this measurement. According to the derivation in Section 2, it is evident that the kick cannot affect the measurement result if the measurement is space-like separated from the kick.

In the Sorkin-type impossible measurements problem, there are additional measurement processes between the kick and the spacelike separated measurement. As mentioned in Section 1, traditional quantum mechanics is governed by two laws: the time evolution of quantum states and measurement theory. According to the traditionally accepted measurement theory, if there are additional measurement processes between the kick and the spacelike separated measurement, the kick has the ability to influence the results of the spacelike separated measurement, which violates relativistic causality. However, within the framework of the one-world interpretation and in combination with the causality demonstrated in Section 2, this troublesome Sorkin-type impossible measurements problem can be resolved. The one-world interpretation holds that quantum mechanics is complete, and thus, the underlying physical laws behind the measurement theory corresponding to this interpretation are still the time evolution of quantum states. Therefore, regardless of whether there are additional measurements between the kick and the spacelike separated measurement, the entire system can always be described using the evolving quantum states over time, where "state update" also falls under the time evolution of quantum states. Consequently, according to the causality demonstrated in Section 2, the kick cannot influence measurements spacelike separated from it under any circumstances. Furthermore in the one-world interpretation, each measurement process has a definite final outcome, and the quantum states encompass all the results of measurements, including those of intermediate measurements. Each measurement result can be read from the reduced density matrix of the region where the measurement occurs. In this sense, the one-world interpretation is a deterministic theory, but surprisingly it neither violates causality nor undermines the completeness of quantum mechanics.

# 5   Conclusion and Outlook

The central result of this paper is that the traditional derivation of the Schrödinger's cat paradox becomes problematic when examined within the framework of relativistic quantum field theory. The conventional derivation of the Schrödinger's cat paradox is based on non-relativistic quantum mechanics. However, the most fundamental and modern framework of quantum theory is relativistic quantum field theory. If one adopts the viewpoint that everything is composed of the same fundamental fields, one finds that the detector and the system being measured are already

entangled even before any measurement takes place, and the operations by which one prepares the microscopic system to be measured are constrained by relativistic causality. These features prevent us from deriving the Schrödinger's cat paradox in quantum field theory in the same way as in the traditional argument. Moreover, we show that the Reeh–Schlieder theorem implies that, even if the final state of the measurement process is a superposition of macroscopically distinct classical configurations, the observed measurement outcome may still be a definite macroscopic state rather than a Schrödinger's cat–like superposition. This further supports the possibility that the Schrödinger's cat paradox may not arise in quantum field theory. If one wished to produce a genuine Schrödinger's cat state, it would require an exceedingly complex experimental setup rather than being achievable by an ordinary measurement.

If the Schrödinger's cat paradox truly does not arise, this suggests a possible pathway toward an interpretation that preserves the completeness of quantum theory while obviating the need for the "many-worlds scenario". For the sake of discussion, we provisionally refer to this interpretation as the "one-world interpretation". In the one-world interpretation, measurement outcomes are deterministic in principle, being fully determined by the initial quantum state of the entire system. However, in practice, we cannot prepare exactly the same microscopic state of every atom and molecule in the detector (including its environment) in each repetition of the experiment. This practical impossibility leads to the apparent randomness of experimental outcomes. Clearly, this interpretation is a kind of hidden variable theory. However, as explained in Section 4, because it violate Statistical Independence (for the meaning of "Statistical Independence", see Ref. [48, 49]), it does not fall under Bell's local hidden variable theory [47] or Leggett's definition of "nonlocal hidden-variable theory" [50]. Therefore, it will not be ruled out by experiments related to Bell's inequality and Leggett's inequality. In fact, the one-world interpretation closely resembles a superdeterministic theory [49]; however, unlike traditional superdeterministic approaches, it maintains that nonlocal quantum mechanics is a complete theory. Combining the causality defined in this paper, it can be concluded that the one-world interpretation is compatible with special relativity, and situations involving superluminal transmission of information do not arise, nor does the Sorkin-type impossible measurements problem.

While our work has the potential to provide a framework that harmoniously integrates relativistic causality, quantum nonlocality, and quantum measurement, we currently still lack a quantitative formulation of the one-world interpretation. Most crucially, we do not yet have a clear understanding of the correspondence between quantum states and macroscopic states, where "macroscopic states" refer to states described in classical concepts, as introduced in Section 4.4. In certain simple cases, it is possible to establish a correspondence between quantum states and macroscopic states [42]. However, for general and complex macroscopic configurations, identifying the corresponding quantum state is often highly nontrivial. For example, it remains extremely difficult to determine the quantum state that would describe a cat.

In addition to clarifying the correspondence between macroscopic states and quantum states, many fundamental questions concerning the measurement process remain to be explored. For example, how can one explicitly write down the complete initial quantum state of a measurement experiment? How can one quantitatively demonstrate that microscopic variations in the initial state are amplified into macroscopic differences during the measurement process? Another important question is how to show, in a quantitative manner, that the final quantum state evolves

into a definite macroscopic state rather than a superposition of macroscopically distinct states. One possible approach is to model the detector as a highly excited bound state in quantum field theory.

While we ultimately aspire to a fully quantum-mechanical detector model, semi-classical detector models (e.g. Refs. [43], [44]) may already help address some of these questions and provide useful guidance for further work. Moreover, the one-world interpretation is a deterministic theory in which tiny variations in the initial state of the detector can produce large changes in individual measurement outcomes, while the statistical distribution of outcomes remains regular. This behavior closely resembles classical chaos; accordingly, it may be fruitful to study the measurement process as a chaotic system.

Finally, it should be clarified that we have not provided a rigorous proof that Schrödinger's cat paradox cannot arise within the framework of quantum field theory (QFT). Therefore, the question of whether the cat paradox can occur in QFT deserves further careful investigation. In fact, the most direct way to prove that the Schrödinger's cat paradox does not arise would be to develop the one-world interpretation in a fully quantitative manner. On the other hand, the simple traditional derivation of the Schrödinger's cat paradox cannot be straightforwardly carried out in quantum field theory. Thus, even an attempt to prove rigorously that the Schrödinger's cat paradox does arise in QFT would ultimately require confronting the same foundational problems that must be addressed when developing the one-world interpretation.

# 6 Acknowledgments

We gratefully acknowledge fruitful conversations with Bartek Czech. We also thank Qi Chen and Weijun Kong for helpful discussions.

# A The derivation of the free field propagator

The propagator for a quantum harmonic oscillator, denoted as $\langle q|\mathrm{e}^{-i\hat{H}_o t}|q_1\rangle$, takes the form $\mathrm{e}^{iS(q,q_1;t)}$, where $S(q, q_1; t)$ is the extreme of the action under fixed path-boundaries (i.e., for fixed $q(0) = q$ and $q(t) = q_1$). The free field theory described by the Hamiltonian $\hat{H} = \int \mathrm{d}^3x \left[\frac{1}{2}\hat{\pi}^2(\boldsymbol{x}) + \frac{1}{2}\left(\nabla\hat{\phi}(\boldsymbol{x})\right)^2 + \frac{1}{2}m^2\hat{\phi}^2(\boldsymbol{x})\right]$ can be regarded as a collection of many harmonic oscillators, therefore we can similarly guess that the field propagator $\langle\phi|\mathrm{e}^{-i\hat{H}t}|\phi_1\rangle$ also takes the same form:

$$\langle\phi|\mathrm{e}^{-i\hat{H}t}|\phi_1\rangle = N(t)\mathrm{e}^{iS(\phi,\phi_1;t)} , \tag{29}$$

where $S(\phi, \phi_1; t)$ is the extremum of the action evaluated for fixed initial state $\phi_1$ and final state $\phi$, and $N(t)$ is independent of $\phi$ and $\phi_1$. In fact, substituting the solutions satisfying the Euler-Lagrange equations into $S = \int \mathrm{d}^4x\, \mathcal{L}(\phi(x), \dot{\phi}(x))$ yields

$$\begin{aligned} S(\phi, \phi_1; t) = &\frac{1}{2}\int \mathrm{d}^3x\mathrm{d}^3y\, G(\boldsymbol{x} - \boldsymbol{y}; t)\Big[\phi(\boldsymbol{x})\phi(\boldsymbol{y}) + \phi_1(\boldsymbol{x})\phi_1(\boldsymbol{y})\Big] \\ &- \int \mathrm{d}^3x\mathrm{d}^3y\, g(\boldsymbol{x} - \boldsymbol{y}; t)\phi(\boldsymbol{x})\phi_1(\boldsymbol{y}) , \end{aligned} \tag{30}$$

where

$$G(\boldsymbol{x} - \boldsymbol{y}; t) = \int \frac{\mathrm{d}^3 p}{(2\pi)^3} \frac{p^0 \cos(p^0 t)}{\sin(p^0 t)} \mathrm{e}^{i\boldsymbol{p}\cdot(\boldsymbol{x}-\boldsymbol{y})} \ ,$$

$$g(\boldsymbol{x} - \boldsymbol{y}; t) = \int \frac{\mathrm{d}^3 p}{(2\pi)^3} \frac{p^0}{\sin(p^0 t)} \mathrm{e}^{i\boldsymbol{p}\cdot(\boldsymbol{x}-\boldsymbol{y})} \ . \tag{31}$$

Eq. (29) is just our conjecture regarding the field propagator $\langle\phi|\mathrm{e}^{-i\hat{H}t}|\phi_1\rangle$. We need to verify that it reduces to $\prod_{x} \delta(\phi(x) - \phi_1(x))$ at $t = 0$, and satisfies the Schrödinger equation for any $t$:

$$i\frac{\partial}{\partial t}\langle\phi|\mathrm{e}^{-i\hat{H}t}|\phi_1\rangle = \hat{H}\langle\phi|\mathrm{e}^{-i\hat{H}t}|\phi_1\rangle \ . \tag{32}$$

Note that in Eq. (32), the first $\hat{H}$ on the right-hand side actually represents the Hamiltonian in the representation defined by the eigenstates of the field operator $\hat{\phi}(x)$. Since the entire derivation involves only one representation, writing it this way does not introduce ambiguity. In the representation defined by the eigenstates of the field operator $\hat{\phi}(x)$, we have $\hat{\phi}(x) = \phi(x)$ and $\hat{\pi}(x) = -i\frac{\delta}{\delta\phi(x)}$.

To handle the Schrödinger equation (32), we need to compute $\int \mathrm{d}^3 x \, \hat{\pi}^2(x)\langle\phi|\mathrm{e}^{-i\hat{H}t}|\phi_1\rangle$. Next, let's proceed to compute it. Firstly, according to Eq. (31), it is straightforward to calculate the specific expressions for $\frac{\partial}{\partial t}G(\boldsymbol{x} - \boldsymbol{y}; t)$ and $\frac{\partial}{\partial t}g(\boldsymbol{x} - \boldsymbol{y}; t)$:

$$\frac{\partial}{\partial t}G(\boldsymbol{x} - \boldsymbol{y}; t) = -\int \frac{\mathrm{d}^3 p}{(2\pi)^3} \frac{(p^0)^2}{\sin^2(p^0 t)} \mathrm{e}^{i\boldsymbol{p}\cdot(\boldsymbol{x}-\boldsymbol{y})} \ ,$$

$$\frac{\partial}{\partial t}g(\boldsymbol{x} - \boldsymbol{y}; t) = -\int \frac{\mathrm{d}^3 p}{(2\pi)^3} \frac{(p^0)^2 \cos(p^0 t)}{\sin^2(p^0 t)} \mathrm{e}^{i\boldsymbol{p}\cdot(\boldsymbol{x}-\boldsymbol{y})} \ . \tag{33}$$

Secondly, utilizing Eq. (33), we can obtain the following formulas:

$$\int \mathrm{d}^3 x \, G(\boldsymbol{x} - \boldsymbol{y}; t)G(\boldsymbol{x} - \boldsymbol{z}; t) = -\frac{\partial}{\partial t}G(\boldsymbol{z} - \boldsymbol{y}; t) + \nabla_z^2 \delta^3(\boldsymbol{z} - \boldsymbol{y}) - m^2 \delta^3(\boldsymbol{z} - \boldsymbol{y}),$$

$$\int \mathrm{d}^3 x \, G(\boldsymbol{x} - \boldsymbol{y}; t)g(\boldsymbol{x} - \boldsymbol{z}; t) = -\frac{\partial}{\partial t}g(\boldsymbol{z} - \boldsymbol{y}; t) \ , \tag{34}$$

$$\int \mathrm{d}^3 x \, g(\boldsymbol{x} - \boldsymbol{y}; t)g(\boldsymbol{x} - \boldsymbol{z}; t) = -\frac{\partial}{\partial t}G(\boldsymbol{z} - \boldsymbol{y}; t) \ .$$

Finally, utilizing (34), the expression for $\int d^3x\, \hat{\pi}^2(x)\langle\phi|e^{-i\hat{H}t}|\phi_1\rangle$ can be obtained as

$$
\int d^3x\, \hat{\pi}^2(x)\langle\phi|e^{-i\hat{H}t}|\phi_1\rangle
$$

$$
= -i\int d^3x\, G(\mathbf{0};t)\langle\phi|e^{-i\hat{H}t}|\phi_1\rangle
$$

$$
+ \left[\int d^3y d^3z\, \phi(\boldsymbol{y})\phi(\boldsymbol{z})\int d^3x\, G(\boldsymbol{x}-\boldsymbol{y};t)G(\boldsymbol{x}-\boldsymbol{z};t)\right.
$$

$$
+ \int d^3y d^3z\, \phi_1(\boldsymbol{y})\phi_1(\boldsymbol{z})\int d^3x\, g(\boldsymbol{x}-\boldsymbol{y};t)g(\boldsymbol{x}-\boldsymbol{z};t)
$$

$$
- \int d^3y d^3z\, \phi_1(\boldsymbol{y})\phi(\boldsymbol{z})\int d^3x\, g(\boldsymbol{x}-\boldsymbol{y};t)G(\boldsymbol{x}-\boldsymbol{z};t)
$$

$$
\left. - \int d^3y d^3z\, \phi(\boldsymbol{y})\phi_1(\boldsymbol{z})\int d^3x\, G(\boldsymbol{x}-\boldsymbol{y};t)g(\boldsymbol{x}-\boldsymbol{z};t)\right]\langle\phi|e^{-i\hat{H}t}|\phi_1\rangle
\tag{35}
$$

$$
= -i\int d^3x\, G(\mathbf{0};t)\langle\phi|e^{-i\hat{H}t}|\phi_1\rangle
$$

$$
- \left[\int d^3x\, (\nabla\phi(x))^2 + m^2\int d^3x\, \phi^2(x)\right]\langle\phi|e^{-i\hat{H}t}|\phi_1\rangle
$$

$$
- \frac{\partial}{\partial t}\left[-2\int d^3x d^3y\, g(\boldsymbol{x}-\boldsymbol{y};t)\phi(\boldsymbol{x})\phi_1(\boldsymbol{y})\right.
$$

$$
\left. + \int d^3x d^3y\, G(\boldsymbol{x}-\boldsymbol{y};t)\Big[\phi(\boldsymbol{x})\phi(\boldsymbol{y}) + \phi_1(\boldsymbol{x})\phi_1(\boldsymbol{y})\Big]\right]\langle\phi|e^{-i\hat{H}t}|\phi_1\rangle\,.
$$

With Eq. (35), we immediately know that the expression on the right-hand side of the Schrödinger equation (32) is given by

$$
\hat{H}\langle\phi|e^{-i\hat{H}t}|\phi_1\rangle
$$

$$
= -i\frac{1}{2}\int d^3x\, G(\mathbf{0};t)\langle\phi|e^{-i\hat{H}t}|\phi_1\rangle
$$

$$
- \frac{1}{2}\frac{\partial}{\partial t}\left[-2\int d^3x d^3y\, g(\boldsymbol{x}-\boldsymbol{y};t)\phi(\boldsymbol{x})\phi_1(\boldsymbol{y})\right.
\tag{36}
$$

$$
\left. + \int d^3x d^3y\, G(\boldsymbol{x}-\boldsymbol{y};t)\Big[\phi(\boldsymbol{x})\phi(\boldsymbol{y}) + \phi_1(\boldsymbol{x})\phi_1(\boldsymbol{y})\Big]\right]\langle\phi|e^{-i\hat{H}t}|\phi_1\rangle\,.
$$

The left-hand side of the Schrödinger equation (32) can be directly computed as

$$
i\frac{\partial}{\partial t}\langle\phi|\mathrm{e}^{-i\hat{H}t}|\phi_1\rangle
$$
$$
=i\frac{1}{N(t)}\frac{dN(t)}{dt}\langle\phi|\mathrm{e}^{-i\hat{H}t}|\phi_1\rangle
$$
$$
-\frac{1}{2}\frac{\partial}{\partial t}\Bigg[-2\int \mathrm{d}^3x\mathrm{d}^3y\ g(\boldsymbol{x}-\boldsymbol{y};t)\phi(\boldsymbol{x})\phi_1(\boldsymbol{y})
$$
$$
+\int \mathrm{d}^3x\mathrm{d}^3y\ G(\boldsymbol{x}-\boldsymbol{y};t)\Big[\phi(\boldsymbol{x})\phi(\boldsymbol{y})+\phi_1(\boldsymbol{x})\phi_1(\boldsymbol{y})\Big]\Bigg]\langle\phi|\mathrm{e}^{-i\hat{H}t}|\phi_1\rangle\ . \tag{37}
$$

Substituting the specific expressions from Eq. (36) and Eq. (37) into the Schrödinger equation (32), we obtain an equation involving $N(t)$:

$$
\frac{d}{dt}\ln N(t) = -\frac{1}{2}\int \mathrm{d}^3x\ G(\boldsymbol{0};t)\ . \tag{38}
$$

This is consistent with the earlier assumption that $N(t)$ is independent of $\phi$ and $\phi_1$, indicating that our previous conjecture Eq. (29) indeed satisfies the Schrödinger equation, and the specific expression for $N(t)$ is

$$
N(t) = \mathcal{N}\mathrm{e}^{-\frac{1}{2}\int \mathrm{d}^3x\int dt\ G(\boldsymbol{0};t)}\ . \tag{39}
$$

However, merely demonstrating that (29) satisfies the Schrödinger equation as a solution is not sufficient. We also need to examine the behavior of $\langle\phi|\mathrm{e}^{-i\hat{H}t}|\phi_1\rangle$ as $t\to 0$. Utilizing (39) and (30), we can express (29) in a more specific form:

$$
\langle\phi|\mathrm{e}^{-i\hat{H}t}|\phi_1\rangle
$$
$$
= \mathcal{N}\mathrm{e}^{-\frac{1}{2}\int \mathrm{d}^3x\int dt\ G(\boldsymbol{0};t)}\exp\Bigg\{-i\int \mathrm{d}^3x\mathrm{d}^3y\ g(\boldsymbol{x}-\boldsymbol{y};t)\phi(\boldsymbol{x})\phi_1(\boldsymbol{y})
$$
$$
+\frac{i}{2}\int \mathrm{d}^3x\mathrm{d}^3y\ G(\boldsymbol{x}-\boldsymbol{y};t)\Big[\phi(\boldsymbol{x})\phi(\boldsymbol{y})+\phi_1(\boldsymbol{x})\phi_1(\boldsymbol{y})\Big]\Bigg\}\ . \tag{40}
$$

As $t\to 0$, it is easy to obtain from (31) that $G(\boldsymbol{x}-\boldsymbol{y};t)\to \frac{1}{t}\delta^3(\boldsymbol{x}-\boldsymbol{y})$ and $g(\boldsymbol{x}-\boldsymbol{y};t)\to \frac{1}{t}\delta^3(\boldsymbol{x}-\boldsymbol{y})$. Consequently, the behavior of (40) as $t\to 0$ can be obtained:

$$
\lim_{t\to 0}\langle\phi|\mathrm{e}^{-i\hat{H}t}|\phi_1\rangle
$$
$$
= \mathcal{N}\lim_{t\to 0}\mathrm{e}^{-\frac{1}{2}\int \mathrm{d}^3x\delta^3(\boldsymbol{0})\ln(t)}\mathrm{e}^{\frac{i}{2t}\int \mathrm{d}^3x[\phi(\boldsymbol{x})-\phi_1(\boldsymbol{x})]^2}\ . \tag{41}
$$

Note that after regularization, the term $\delta^3(\boldsymbol{0})$ effectively becomes $\frac{1}{\mathrm{d}x^3}$. Expanding the integral in the exponential of Eq. (41) into a product of exponentials, we obtain:

$$
\lim_{t\to 0}\langle\phi|\mathrm{e}^{-i\hat{H}t}|\phi_1\rangle = \mathcal{N}\lim_{t\to 0}\prod_x\frac{1}{\sqrt{t}}\mathrm{e}^{\frac{i(\mathrm{d}x)^3}{2t}[\phi(\boldsymbol{x})-\phi_1(\boldsymbol{x})]^2} = \prod_x\delta(\phi(x)-\phi_1(x))\ , \tag{42}
$$

where $\mathcal{N} \equiv \prod_x \left( \frac{(\mathrm{d}x)^3}{2\pi i} \right)$. This implies that (40) not only serves as a solution to the Schrödinger equation, but also represents the inner product between field operator eigenstates as $t \rightarrow 0$, indicating that (40) and (29) are the correct expressions for the field propagator.

# B  A detailed proof of Eq. (7)

Dividing the space into two parts, denoted as $B$ and $b$, with the values of $\phi$ on the two regions denoted as $\phi_B$ and $\phi_b$, the reduced density matrix of the region $B$ in Fig. 1 is given by

$$\int \mathcal{D}\phi_b \, \rho(\phi, \phi'; t)\bigg|_{\phi_b=\phi'_b} = |N(t)|^2 \int \mathcal{D}\varphi \mathcal{D}\varphi' \rho(\varphi, \varphi') \int \mathcal{D}\phi_b \, K(\phi, \phi', \varphi, \varphi'; t)\bigg|_{\phi_b=\phi'_b} . \tag{43}$$

Further utilization of Eq. (4) and Eq. (5) yields the final integral of Eq. (43) as

$$\int \mathcal{D}\phi_b \, K(\phi, \phi', \varphi, \varphi'; t)\bigg|_{\phi_b=\phi'_b}$$

$$= \exp\left\{ \frac{i}{2} \int_B \mathrm{d}^3x \int_B \mathrm{d}^3y \, G(\boldsymbol{x} - \boldsymbol{y}; t) \Big[ \phi_B(\boldsymbol{x})\phi_B(\boldsymbol{y}) - \phi'_B(\boldsymbol{x})\phi'_B(\boldsymbol{y}) \Big] \right.$$

$$+ \frac{i}{2} \int \mathrm{d}^3x \int \mathrm{d}^3y \, G(\boldsymbol{x} - \boldsymbol{y}; t) \Big[ \varphi(\boldsymbol{x})\varphi(\boldsymbol{y}) - \varphi'(\boldsymbol{x})\varphi'(\boldsymbol{y}) \Big] \tag{44}$$

$$\left. - i \int_B \mathrm{d}^3x \int \mathrm{d}^3y \, g(\boldsymbol{x} - \boldsymbol{y}; t) \Big[ \phi_B(\boldsymbol{x})\varphi(\boldsymbol{y}) - \phi'_B(\boldsymbol{x})\varphi'(\boldsymbol{y}) \Big] \right\}$$

$$\times \prod_{\boldsymbol{x}\in b} \delta\left[ \int_B \mathrm{d}^3y \, G(\boldsymbol{x} - \boldsymbol{y}; t) \, [\phi_B(\boldsymbol{y}) - \phi'_B(\boldsymbol{y})] - \int \mathrm{d}^3y \, g(\boldsymbol{x} - \boldsymbol{y}; t) \, [\varphi(\boldsymbol{y}) - \varphi'(\boldsymbol{y})] \right] .$$

According to Eq. (6), the inverse of the function $g(\boldsymbol{x} - \boldsymbol{y}; t)$ can be determined as

$$g_{-1}(\boldsymbol{x} - \boldsymbol{y}; t) = \int \frac{d^3p}{(2\pi)^3} \frac{\sin(p^0 t)}{p^0} e^{i\boldsymbol{p}\cdot(\boldsymbol{x}-\boldsymbol{y})} . \tag{45}$$

It is easy to verify that the functions $g(\boldsymbol{x} - \boldsymbol{y}; t)$ and $g_{-1}(\boldsymbol{x} - \boldsymbol{y}; t)$ satisfy

$$\int \mathrm{d}^3y \, g_{-1}(\boldsymbol{x} - \boldsymbol{y}; t) g(\boldsymbol{y} - \boldsymbol{z}; t) = \delta(\boldsymbol{x} - \boldsymbol{z}) . \tag{46}$$

Based on Eq. (6) and Eq. (45), we can also obtain the following useful formulas:

$$\int \mathrm{d}^3y \, g_{-1}(\boldsymbol{x} - \boldsymbol{y}; t) G(\boldsymbol{y} - \boldsymbol{z}; t) = \frac{\partial}{\partial t} g_{-1}(\boldsymbol{x} - \boldsymbol{z}; t) , \tag{47}$$

$$\int \mathrm{d}^3z \, \frac{\partial}{\partial t} g_{-1}(\boldsymbol{x} - \boldsymbol{z}; t) G(\boldsymbol{z} - \boldsymbol{y}; t) = g(\boldsymbol{x} - \boldsymbol{y}; t) + \frac{\partial^2}{\partial t^2} g_{-1}(\boldsymbol{x} - \boldsymbol{y}; t) . \tag{48}$$

Based on Eq. (45), we also note an important equality: $g_{-1}(\boldsymbol{x} - \boldsymbol{y}; t) = \left[ \hat{\phi}(\boldsymbol{x}, t), \hat{\phi}(\boldsymbol{y}, 0) \right]$, where $\hat{\phi}(\boldsymbol{x}, t)$ is the field operator in the Heisenberg picture. Therefore, it is evident that

$$g_{-1}(\boldsymbol{x}; t) = 0 , \quad \boldsymbol{x}^2 - t^2 > 0 . \tag{49}$$

As shown in Fig. 1, the region $a$ is the $t = 0$ surface in the past domain of dependence of region $b$. Combining Fig. 1 with Eq. (49), we can rewrite Eq. (46), Eq. (47), and Eq. (48) as follows:

$$\int_b \mathrm{d}^3 y \; g_{-1}(\boldsymbol{x} - \boldsymbol{y}; t) g(\boldsymbol{y} - \boldsymbol{z}; t) = \delta(\boldsymbol{x} - \boldsymbol{z}), \; \boldsymbol{x} \in a, \tag{50}$$

$$\int_b \mathrm{d}^3 y \; g_{-1}(\boldsymbol{x} - \boldsymbol{y}; t) G(\boldsymbol{y} - \boldsymbol{z}; t) = \frac{\partial}{\partial t} g_{-1}(\boldsymbol{x} - \boldsymbol{z}; t), \; \boldsymbol{x} \in a, \tag{51}$$

$$\int_b \mathrm{d}^3 z \; \frac{\partial}{\partial t} g_{-1}(\boldsymbol{x} - \boldsymbol{z}; t) G(\boldsymbol{z} - \boldsymbol{y}; t) = g(\boldsymbol{x} - \boldsymbol{y}; t) + \frac{\partial^2}{\partial t^2} g_{-1}(\boldsymbol{x} - \boldsymbol{y}; t), \; \boldsymbol{x} \in a. \tag{52}$$

Let's focus back on Eq. (44). The $\delta$ function in Eq. (44) indicates that

$$\int \mathrm{d}^3 y \; g(\boldsymbol{x} - \boldsymbol{y}; t) \left[ \varphi(\boldsymbol{y}) - \varphi'(\boldsymbol{y}) \right] = \int_B \mathrm{d}^3 y \; G(\boldsymbol{x} - \boldsymbol{y}; t) \left[ \phi_B(\boldsymbol{y}) - \phi'_B(\boldsymbol{y}) \right], \; \boldsymbol{x} \in b. \tag{53}$$

Based on Eq. (50), Eq. (53), Eq. (51), and Eq. (49), in region $a$ we can obtain

$$
\begin{aligned}
& \varphi_a(\boldsymbol{x}) - \varphi'_a(\boldsymbol{x}) \\
&= \int \mathrm{d}^3 y \; \delta(\boldsymbol{x} - \boldsymbol{y}) \left[ \varphi_a(\boldsymbol{y}) - \varphi'_a(\boldsymbol{y}) \right] \\
&= \int \mathrm{d}^3 y \int_b \mathrm{d}^3 z \; g_{-1}(\boldsymbol{x} - \boldsymbol{z}; t) g(\boldsymbol{z} - \boldsymbol{y}; t) \left[ \varphi_a(\boldsymbol{y}) - \varphi'_a(\boldsymbol{y}) \right] \\
&= \int_B \mathrm{d}^3 y \int_b \mathrm{d}^3 z \; g_{-1}(\boldsymbol{x} - \boldsymbol{z}; t) G(\boldsymbol{z} - \boldsymbol{y}; t) \left[ \phi_B(\boldsymbol{y}) - \phi'_B(\boldsymbol{y}) \right] \\
&= \int_B \mathrm{d}^3 y \int_b \mathrm{d}^3 z \; g_{-1}(\boldsymbol{x} - \boldsymbol{z}; t) G(\boldsymbol{z} - \boldsymbol{y}; t) \left[ \phi_B(\boldsymbol{y}) - \phi'_B(\boldsymbol{y}) \right] \\
&= 0 \,, \qquad\qquad\qquad \boldsymbol{x} \in a,
\end{aligned}
\tag{54}
$$

where, the second equality employs Eq. (50), the third equality employs Eq. (53), the fourth equality employs Eq. (51), and the final equality employs Eq. (49). Then, according to Eq. (54), the $\delta$ function in (44) can be expressed as

$$
\begin{aligned}
\propto & \prod_{\boldsymbol{x} \in a} \delta \Big( \varphi_a(\boldsymbol{x}) - \varphi'_a(\boldsymbol{x}) \Big) \\
& \times \prod_{\boldsymbol{x} \in b} \delta \bigg( \int_B \mathrm{d}^3 y \; G(\boldsymbol{x} - \boldsymbol{y}; t) \left[ \phi_B(\boldsymbol{y}) - \phi'_B(\boldsymbol{y}) \right] \\
& \qquad\qquad - \int_A \mathrm{d}^3 y \; g(\boldsymbol{x} - \boldsymbol{y}; t) \left[ \varphi_A(\boldsymbol{y}) - \varphi'_A(\boldsymbol{y}) \right] \bigg) \,.
\end{aligned}
\tag{55}
$$

The proportionality coefficient in Eq. (55) is independent of $\varphi_a$ and $\varphi'_a$. Substituting Eq. (55) into Eq. (44), we obtain

$$\int \mathcal{D}\phi_b \, K(\phi, \phi', \varphi, \varphi'; t)\Big|_{\phi_b = \phi'_b}$$

$$\propto \exp\left\{ i \int_a \mathrm{d}^3 x \, \varphi_a(\boldsymbol{x}) \left[ \int_A \mathrm{d}^3 y \, G(\boldsymbol{x} - \boldsymbol{y}; t) \left[ \varphi_A(\boldsymbol{y}) - \varphi'_A(\boldsymbol{y}) \right] \right.\right.$$

$$\left.\left. - \int_B \mathrm{d}^3 y \, g(\boldsymbol{x} - \boldsymbol{y}; t) \left[ \phi_B(\boldsymbol{y}) - \phi'_B(\boldsymbol{y}) \right] \right] \right\}$$

$$\times \exp\left\{ \frac{i}{2} \int_B \mathrm{d}^3 x \int_B \mathrm{d}^3 y \, G(\boldsymbol{x} - \boldsymbol{y}; t) \left[ \phi_B(\boldsymbol{x})\phi_B(\boldsymbol{y}) - \phi'_B(\boldsymbol{x})\phi'_B(\boldsymbol{y}) \right] \right.$$

$$+ \frac{i}{2} \int_A \mathrm{d}^3 x \int_A \mathrm{d}^3 y \, G(\boldsymbol{x} - \boldsymbol{y}; t) \left[ \varphi_A(\boldsymbol{x})\varphi_A(\boldsymbol{y}) - \varphi'_A(\boldsymbol{x})\varphi'_A(\boldsymbol{y}) \right]$$

$$\left. - i \int_A \mathrm{d}^3 x \int_B \mathrm{d}^3 y \, g(\boldsymbol{x} - \boldsymbol{y}; t) \left[ \varphi_A(\boldsymbol{x})\phi_B(\boldsymbol{y}) - \varphi'_A(\boldsymbol{x})\phi'_B(\boldsymbol{y}) \right] \right\}$$

$$\times \prod_{\boldsymbol{x} \in b} \delta\left( \int_B \mathrm{d}^3 y \, G(\boldsymbol{x} - \boldsymbol{y}; t) \left[ \phi_B(\boldsymbol{y}) - \phi'_B(\boldsymbol{y}) \right] - \int_A \mathrm{d}^3 y \, g(\boldsymbol{x} - \boldsymbol{y}; t) \left[ \varphi_A(\boldsymbol{y}) - \varphi'_A(\boldsymbol{y}) \right] \right)$$

$$\times \prod_{\boldsymbol{x} \in a} \delta\left( \varphi_a(\boldsymbol{x}) - \varphi'_a(\boldsymbol{x}) \right). \tag{56}$$

Next, we proceed to prove that the first two lines (the first exponential term) of Eq. (56) equals 1. Note that the first $\delta$ function in Eq. (56) implies

$$\int_B \mathrm{d}^3 y \, G(\boldsymbol{x} - \boldsymbol{y}; t) \left[ \phi_B(\boldsymbol{y}) - \phi'_B(\boldsymbol{y}) \right] = \int_A \mathrm{d}^3 y \, g(\boldsymbol{x} - \boldsymbol{y}; t) \left[ \varphi_A(\boldsymbol{y}) - \varphi'_A(\boldsymbol{y}) \right], \ \boldsymbol{x} \in b. \tag{57}$$

According to Eq. (50), Eq. (51), Eq. (49), and Eq. (57), we can derive the following expression for $\boldsymbol{x} \in a$:

$$\int_A \mathrm{d}^3 y \, G(\boldsymbol{x} - \boldsymbol{y}; t) \left[ \varphi_A(\boldsymbol{y}) - \varphi'_A(\boldsymbol{y}) \right]$$

$$= \int_A \mathrm{d}^3 y \int \mathrm{d}^3 s \, G(\boldsymbol{x} - \boldsymbol{s}; t) \delta(\boldsymbol{s} - \boldsymbol{y}) \left[ \varphi_A(\boldsymbol{y}) - \varphi'_A(\boldsymbol{y}) \right]$$

$$= \int \mathrm{d}^3 z \int \mathrm{d}^3 s \, G(\boldsymbol{x} - \boldsymbol{s}; t) g_{-1}(\boldsymbol{s} - \boldsymbol{z}; t) \int_A \mathrm{d}^3 y \, g(\boldsymbol{z} - \boldsymbol{y}; t) \left[ \varphi_A(\boldsymbol{y}) - \varphi'_A(\boldsymbol{y}) \right]$$

$$= \int \mathrm{d}^3 z \, \frac{\partial}{\partial t} g_{-1}(\boldsymbol{x} - \boldsymbol{z}; t) \int_A \mathrm{d}^3 y \, g(\boldsymbol{z} - \boldsymbol{y}; t) \left[ \varphi_A(\boldsymbol{y}) - \varphi'_A(\boldsymbol{y}) \right]$$

$$= \int_b \mathrm{d}^3 z \, \frac{\partial}{\partial t} g_{-1}(\boldsymbol{x} - \boldsymbol{z}; t) \int_A \mathrm{d}^3 y \, g(\boldsymbol{z} - \boldsymbol{y}; t) \left[ \varphi_A(\boldsymbol{y}) - \varphi'_A(\boldsymbol{y}) \right]$$

$$= \int_b \mathrm{d}^3 z \, \frac{\partial}{\partial t} g_{-1}(\boldsymbol{x} - \boldsymbol{z}; t) \int_B \mathrm{d}^3 y \, G(\boldsymbol{x} - \boldsymbol{y}; t) \left[ \phi_B(\boldsymbol{y}) - \phi'_B(\boldsymbol{y}) \right], \tag{58}$$

where, the second equality uses Eq. (50), the third equality uses Eq. (51), the fourth equality uses Eq. (49), and the final equality uses Eq. (57). Furthermore, utilizing Eq. (52), Eq. (58) can

be written as

$$
\int_A \mathrm{d}^3y \, G(\boldsymbol{x} - \boldsymbol{y}; t) \Big[ \varphi_A(\boldsymbol{y}) - \varphi'_A(\boldsymbol{y}) \Big] - \int_B \mathrm{d}^3y \, g(\boldsymbol{x} - \boldsymbol{y}; t) \Big[ \phi_B(\boldsymbol{y}) - \phi'_B(\boldsymbol{y}) \Big]
$$
$$
= \int_B \mathrm{d}^3y \, \frac{\partial^2}{\partial t^2} g_{-1}(\boldsymbol{x} - \boldsymbol{y}; t) \Big[ \phi_B(\boldsymbol{y}) - \phi'_B(\boldsymbol{y}) \Big] \tag{59}
$$
$$
= 0 \, ,
$$

where $\boldsymbol{x} \in a$ and the last equality utilizes Eq. (49). From Eq. (59), it follows that the first two lines (the first exponential term) of Eq.(56) equals 1. Therefore, utilizing Eq. (59) and Eq. (56), Eq. (43) can be expressed in the following form:

$$
\int \mathcal{D}\phi_b \, \rho(\phi, \phi'; t) \bigg|_{\phi_b = \phi'_b} = \int \mathcal{D}\varphi_A \mathcal{D}\varphi'_A F(\phi_B, \phi'_B, \varphi_A, \varphi'_A; t) \int \mathcal{D}\varphi_a \, \rho(\varphi, \varphi') \bigg|_{\varphi_a = \varphi'_a} . \tag{60}
$$

This is precisely Eq. (7).

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
