# Peer review of "Causality and the Interpretation of Quantum Mechanics"

_SciPost Physics_

## Round 2 · Referee Report · Anonymous (Referee 1) · 2025-10-31

Strengths

This paper contains valuable observations that should be disseminated. In particular, it aims to show that the Schroedinger cat paradox goes away if one follows the lessons of relativistically covariant quantum field theory, including thinking carefully about quantum field entanglement. The discussion of apparatus details as hidden variables is especially insightful.

Weaknesses

Broadly speaking, the paper is divided into three major segments. The first, Section 2 but excluding Section 2.2, demonstrates that a free scalar quantum field theory obeys causality in the sense of propagating localized states in time in accordance with the familiar geometry of future light cones. This is intuitively obvious from Lorentz invariance, which even renormalizable interacting quantum fields respect, but perhaps the derivation in terms of local field operators, rather than particle modes, is novel. I think the authors’ purpose in this major segment is to set the stage for the second major segment (see below), but it’s not clear to me why that’s really necessary. In particular, the first major segment deals with two sets of regions, separated by some time interval, while the second major segment seems really to focus only on regions at a single time. In any case, this first segment is mostly an extremely detailed Green’s function calculation that distracts a reader from understanding where the paper is going, and maybe even keeps a reader from having the patience to read further.
The second major segment, Sections 2.2 and 3, demonstrates that any two states whose density matrices agree outside some spacelike reference region, are related to one another by a unitary operation built from fields that live only in the region in question. The point seems to be that this establishes that any two localized states can be transformed into one another by a physical operation that requires no recourse to things like measurement axioms or wavefunction collapse. I have a hard time understanding why the existence of a unitary transformation guarantees that the transformation in question corresponds to an actual physical process. In fact, as developed in Section 2.2, the unitary transformation depends very specifically on the two states in question and is unrelated to any dynamical laws (in particular, the Hamiltonian) that govern evolution in time.
The third and last major segment, Section 4, aims to show that the Schroedinger cat paradox goes away if you follow the lessons of the first two major segments, and think carefully about entanglement in quantum field theory. I think this last segment is quite strong and valuable, but it really seems to not require the material in the first major segment.

Report

The paper is valuable but too much of it is dense symbolic calculation that distracts from any sense of narrative direction. The paper also makes a claim about the physical significance of a certain formal construction that I do not find convincing (I am happy for the authors to set me straight). I feel positively about the journal's acceptance criteria if the paper were rewritten per the recommendations below.

Requested changes

1. Eliminate Section 2 (except for Section 2.2), or move it to an appendix. If it’s not eliminated but moved to an appendix, then include some language up front to explain why the material is even in this paper and how it strengthens Section 4.
2. In Sections 2.2 and 3, explain why the unitary operations constructed formally in Section 2.2 have any physical significance. If the authors can’t explain this adequately, then rewrite Section 4 to do without reference to such operations.

Recommendation

Ask for major revision

  • validity: good
  • significance: high
  • originality: high
  • clarity: high
  • formatting: excellent
  • grammar: excellent

Author:  Kaixun Tu  on 2025-11-04  [id 5992]

(in reply to Report 1 on 2025-10-31)

The referee writes:

Broadly speaking, the paper is divided into three major segments. The first, Section 2 but excluding Section 2.2, demonstrates that a free scalar quantum field theory obeys causality in the sense of propagating localized states in time in accordance with the familiar geometry of future light cones. This is intuitively obvious from Lorentz invariance, which even renormalizable interacting quantum fields respect, but perhaps the derivation in terms of local field operators, rather than particle modes, is novel. I think the authors’ purpose in this major segment is to set the stage for the second major segment (see below), but it’s not clear to me why that’s really necessary.

Our response 1:

Let us explain why we devoted significant effort to writing the first major segment. In brief, the main purpose of this segment is to help a broad readership from different fields clearly understand what we mean by causality as defined in our work.

As stated in Section 1 of this paper, there are various definitions of causality in quantum mechanics, stemming from the fact that the notion of “localization” is sometimes ambiguous in quantum theory. In fact, we believe that the causality described in algebraic quantum field theory (AQFT) represents the most advanced and rigorous formulation. However, AQFT itself is too abstract for most readers. Therefore, we propose defining causality in quantum field theory using reduced density matrices, which are a very basic and familiar concept in quantum mechanics. Although our definition of causality is physically equivalent to that in AQFT, it is indeed less mathematically rigorous. For this reason, we provide a concrete and detailed derivation to demonstrate that—even without the full mathematical sophistication of AQFT—the reduced density matrix is still a practically useful object. (In fact, the level of rigor in defining causality through reduced density matrices is comparable to that of the path integral formulations commonly presented in standard quantum field theory textbooks, such as Weinberg’s The Quantum Theory of Fields, Vol. I, Section 9.2, where the vacuum wave functional is derived.)

On the other hand, although the concept of the reduced density matrix is very common and fundamental in conventional quantum mechanics, it is not so in quantum field theory (QFT). Even among QFT researchers, many are more familiar with the Feynman diagram language and the Fock-space representation (i.e., “particle modes”), and rarely work directly with reduced density matrices in QFT. However, when discussing causality and, later on, spatial entanglement in quantum field theory, we cannot rely on the “particle mode” representation — instead, we need use the representation expanded by the eigenstates of local field operators. For a broader audience — especially those who are not QFT specialists but are interested in the foundations of quantum theory — the notion of a reduced density matrix in QFT is even more obscure. For instance, they might naturally ask: What does it mean to take the trace over a spatial region? Therefore, we believe it is necessary to explicitly and completely demonstrate our definition of causality based on reduced density matrices, using the simplest possible QFT model and providing a detailed derivation. (Of course, for researchers studying entanglement entropy in quantum field theory, this major segment is indeed entirely redundant, since they are already very familiar with the concept of reduced density matrices in QFT.)

The referee writes:

  1. Eliminate Section 2 (except for Section 2.2), or move it to an appendix. If it’s not eliminated but moved to an appendix, then include some language up front to explain why the material is even in this paper and how it strengthens Section 4.

Our response 2:

We agree to move the content of Section 2.1 to the Appendix. Our response 1 has already explained why the content of Section 2.1 should remain in this paper. The main purpose of Section 2.1 is to help readers from various fields understand what exactly we mean by causality as defined in our work. Therefore, asking how it strengthens Section 4 is essentially equivalent to asking how causality strengthens Section 4. This question is addressed in Our response 8, where we explain how causality strengthens Section 4.

The referee writes:

In particular, the first major segment deals with two sets of regions, separated by some time interval, while the second major segment seems really to focus only on regions at a single time.

Our response 3:

In fact, the ultimate goal of Section 2.2 in the second major segment is also related to two sets of regions separated by some time interval.

Previous studies in the literature have already shown that: If there exists a unitary operator $U_a$ supported in region a such that $|\psi_2\rangle= U_a |\psi_1\rangle$ and the relationship between regions a and b is as shown in Fig. 1, then we have $tr_b(|\psi_1;t\rangle\langle \psi_1;t|) = tr_b(|\psi_2;t\rangle\langle\psi_2;t|)$. Our goal is to establish a form of causality stating that: $tr_a(|\psi_1\rangle\langle\psi_1|) = tr_a(|\psi_2\rangle\langle\psi_2|) \qquad \Rightarrow\qquad tr_b(|\psi_1;t\rangle\langle\psi_1;t|) = tr_b(|\psi_2;t\rangle\langle\psi_2;t|)$ Therefore, to prove this causality (which indeed concerns “two sets of regions separated by some time interval”), it is sufficient to focus only on regions at a single time and show that: For any two states satisfying $tr_a(|\psi_1\rangle\langle\psi_1|) = tr_a(|\psi_2\rangle\langle\psi_2|)$, there exists a unitary operator $U_a$ such that $|\psi_2\rangle= U_a|\psi_1\rangle$.

The referee writes:

In any case, this first segment is mostly an extremely detailed Green’s function calculation that distracts a reader from understanding where the paper is going, and maybe even keeps a reader from having the patience to read further.

Our response 4:

We sincerely thank the referee for pointing out this issue from the reader’s perspective. We will move this part of the content to the Appendix, so that interested readers can refer to it there.

The referee writes:

The second major segment, Sections 2.2 and 3, demonstrates that any two states whose density matrices agree outside some spacelike reference region, are related to one another by a unitary operation built from fields that live only in the region in question.

Our response 5:

The main derivation in Section 2.2 is indeed as the referee described. However, Section 3 differs somewhat from Section 2.2. The purpose of Section 3 is to argue that a local physical operation must be equivalent to a local unitary operator.

The referee writes:

The point seems to be that this establishes that any two localized states can be transformed into one another by a physical operation that requires no recourse to things like measurement axioms or wavefunction collapse.

Our response 6:

In fact, we do not address the measurement process here; rather, we are arguing that causality imposes constraints on physical operations that can be realized in the real world. The “apparatus (or a human)” mentioned in Section 3 refers to the device that generates the physical operation. If we connect this with the later discussion, the “apparatus (or a human)” in Section 3 is not a measuring device, but rather the one responsible for preparing the initial state of the microscopic system to be measured. For example, in the latter part of Section 4.1, we present an example involving coherent states. One can use an “apparatus (or a human)” to generate a small coherent state $|s_1\rangle$ or $|s_2\rangle$ from the vacuum, but it is impossible to use any “apparatus (or a human)” to create a superposition state $|\psi_3\rangle=|s_1\rangle+|s_2\rangle$ from the vacuum, since such a process would violate causality. Here, $|s_1\rangle$, $|s_2\rangle$, and $|\psi_3\rangle$ represent the microscopic systems being measured, and no measurement has yet taken place. Thus, the “apparatus (or a human)” referred to in Section 3 denotes the means by which one prepares the initial states in an experiment, not the measuring instruments. Devices used for measurement are explicitly called “detectors” in our paper, not “apparatus”.

The referee writes:

I have a hard time understanding why the existence of a unitary transformation guarantees that the transformation in question corresponds to an actual physical process. In fact, as developed in Section 2.2, the unitary transformation depends very specifically on the two states in question and is unrelated to any dynamical laws (in particular, the Hamiltonian) that govern evolution in time.

and

In Sections 2.2 and 3, explain why the unitary operations constructed formally in Section 2.2 have any physical significance. If the authors can’t explain this adequately, then rewrite Section 4 to do without reference to such operations.

Our response 7:

We do not claim that “the existence of a unitary transformation guarantees that the transformation in question corresponds to an actual physical process”. Instead, we argue that a physical process is governed by time evolution under a Hamiltonian. The “unitary operations constructed formally in Section 2.2” have no physical significance. As explained in Our response 3 , they serve merely as an intermediate mathematical tool used to prove causality. In the later sections, we only need to use the resulting conclusion: “any two states whose density matrices agree outside some spacelike reference region, are related to one another by a unitary operation built from fields that live only in the region” . The explicit form of that unitary operator is irrelevant. Section 3 aims to convey a single point: a physical operation localized in a given region must be equivalent to a unitary operator localized within that region. In Section 4, we simply restate this result in another form: a transformation that cannot be written as a localized unitary operator cannot correspond to a physical operation. We do not need to determine what specific transformations qualify as physical operations. As illustrated by the coherent-state example mentioned earlier, one can use a physical operation to create two distinct coherent states from the vacuum. However, it is impossible to produce their superposition state through any physical operation, because the superposition state and the vacuum cannot be connected by a localized unitary transformation.

To better clarify our point, let us briefly illustrate the role of Section 3 in the context of the Schrödinger’s cat paradox. In conventional nonrelativistic quantum mechanics, we usually assume that the microscopic system S and the detector D are not entangled initially, i.e., the joint initial state is $|s_1\rangle|d\rangle$. In that case, we can apply a unitary transformation localized on the microscopic system S to transform $|s_1\rangle|d\rangle$ into $|s_2\rangle|d\rangle$, obtaining the initial state for another experiment. Similarly, we can use a local unitary operation to obtain $|\psi_3\rangle=(|s_1\rangle+|s_2\rangle)|d\rangle$, whose time evolution leads to the Schrödinger’s cat state. However, all of this relies on the assumption that the initial state of the system and detector is not entangled. Section 4.1 aims to show that, when the initial state is entangled, we can no longer use a local physical operation (since no corresponding unitary transformation exists) to prepare an initial state that leads to a Schrödinger’s cat state—that is, we cannot act only on the microscopic system S to produce a Schrödinger’s cat state. Therefore, what we need to know is merely which kinds of transformations are not physical operations, rather than what specific transformations are physical operations. Hence, the “unitary operations constructed formally in Section 2.2” are completely unrelated to the discussion in Section 4.

Remark: The content of Section 3 does not discuss the measurement process, but rather the preparation of the initial state for a measurement experiment. The measurement process is simply the ordinary time evolution governed by the Hamiltonian. We do not deny the existence of Schrödinger’s cat states; however, creating such macroscopic superpositions within the framework of quantum field theory is an extremely complex process—it cannot be achieved merely by manipulating the microscopic system being measured in a standard measurement experiment. In other words, generating a Schrödinger’s cat state requires a much more intricate and delicate experimental setup, and it does not belong to the category of simple measurement experiments.

The referee writes:

The third and last major segment, Section 4, aims to show that the Schroedinger cat paradox goes away if you follow the lessons of the first two major segments, and think carefully about entanglement in quantum field theory. I think this last segment is quite strong and valuable, but it really seems to not require the material in the first major segment.

Our response 8:

We admit that the first major segment does not directly contribute to Section 4. However, it still provides indirect support.

Let us briefly outline the overall logical structure of the paper: 1 We define and prove the causality in quantum field theory. 2 We introduce the constraint that this causality imposes on physical operations. 3 Because of the constraint and the inherent entanglement in quantum field theory, it becomes impossible to create a Schrödinger’s cat state in an ordinary measurement experiment. 4 The absence of the Schrödinger’s cat paradox and the causality in quantum field theory motivate us to propose an interpretation of quantum mechanics that considers quantum mechanics to be complete and remains fully compatible with relativity.

From the above discussion, we can see that the causality principle plays two essential roles in our manuscript. First, it constrains the possible operations on the initial state in measurement experiments, thereby preventing the creation of a Schrödinger’s cat state. Second, it contributes to the construction of our interpretation of quantum mechanics—after all, discussions of measurement in history have often involved causality, such as in the original EPR paradox (instantaneous wavefunction collapse) and in the more recent Sorkin-type impossible measurements problem. This shows that causality is, in fact, a central concept in our paper. As explained in Our response 1, the material in the first major segment is included to help readers from various fields who are interested in quantum interpretation understand our definition of causality. (While researchers studying entanglement entropy are already familiar with the reduced density matrix in quantum field theory, scholars from other areas may not be.)

Summary: We sincerely thank the referee for the valuable suggestions, and we are willing to revise the manuscript according to the referee’s requests. Specifically, we will: 1. Move Section 2.1 to the Appendix and include some language to explain why the material is included in this paper. 2. Clarify that “the unitary operations constructed formally in Section 2.2” have no physical significance. Clarify that we do not make use of it in Section 4; instead, what we use is only the conclusion that “a transformation that cannot be written as a localized unitary operator cannot correspond to a physical operation”.

---

## Editorial Decision

in_refereeing